

**An innovative approach to measuring hygroscopic light scattering enhancement**
**using a humidified single-nephelometer system**
Lenka Suchánková[1,2,3], Jakub Ondráček[1], Naděžda Zíková[1], Petr Roztočil[1], Petr Vodička[1], Roman
Prokeš[2,3], Ivan Holoubek[2,3+], Vladimír Ždímal[1]
[1]Institute of Chemical Process Fundamentals of the Czech Academy of Sciences, Prague, 165 00, Czech Republic
[2]Global Change Research Institute of the Czech Academy of Sciences, Brno, 603 00, Czech Republic
[3]RECETOX, Faculty of Science, Masaryk University, Brno, 611 37, Czech Republic
[+] deceased
*Correspondence to*: Lenka Suchánková (suchankova@icpf.cas.cz)
**Keywords**: aerosol light scattering properties, hygroscopicity, suburban environment, climate change
**Abstract:** Most atmospheric aerosol particles are hygroscopic, meaning they absorb water from the surrounding air, altering
their size, shape, overall chemistry, refractive index, and thus light-scattering properties — an effect with important
implications for Earth's radiative balance. The scattering enhancement factor, f(RH), and backscattering enhancement factor,
f(RH)$_{bsp}$, quantify the increase in light scattering under elevated relative humidity (RH). These parameters are typically
measured using two nephelometers operating under dry (RH < 40 %) and humidified (RH > 80 %) conditions, a method prone
to inter-instrument uncertainties. This study presents a novel single-nephelometer system that reduces measurement
uncertainty and studies aerosol hygroscopic behavior in the inadequately represented European urban environment. The system
was deployed at a suburban site in Prague, Suchdol, Czech Republic, from November 2022 to August 2023. Results revealed
low aerosol hygroscopicity, likely due to a well-mixed aerosol population dominated by black and brown carbon. Both
enhancement factors peaked in spring, possibly influenced by favorable conditions for new particle formation and changes in
aerosol composition, size distribution, and meteorological conditions. In contrast, low values in summer reflected a
composition shift toward black carbon-dominated aerosols from traffic emissions, with particle growth being disrupted,
potentially due to the structural compaction of black carbon aggregates under high RH. While f(RH) and f(RH)$_{bsp}$ generally
increased with decreasing concentrations of light-absorbing particles, organic carbon, particularly its most volatile fractions,
significantly enhanced aerosol hygroscopicity in the urban environment. Despite low aerosol hygroscopicity, increased RH
significantly influenced aerosol climate-relevant variables.
**1. Introduction**
Atmospheric aerosols play a critical role in the Earth's energy budget through direct aerosol-radiation interactions (ARI)
by the scattering and absorption of short- and long-wave radiation and indirect aerosol-cloud interactions (ACI) by changes in
the microphysical and radiative properties of clouds, respectively (Boucher, 2015; IPCC, 2021). The Sixth Assessment Report
of the IPCC estimated the total aerosol effective radiative forcing (ERF) to be -1.1 [-1.7 to -0.4] W m$^{-2}$ over 1750-2019 (Foster
et al., 2023). Despite growing research on the aerosol radiative effects (e.g., Toll et al., 2019; Williams et al., 2022; Zhang et
al., 2025 and references herein), aerosol ERF remains the most significant uncertainty in climate models due to the high spatial
and temporal variability of aerosol properties, limited understanding of pre-industrial aerosol conditions, and the indirect
aerosol-induced changes in the atmosphere (Carslaw et al., 2017; Kahn et al., 2023; Watson-Parris and Smith, 2022).

Hygroscopicity, defined as the ability of aerosol particles to attract and absorb moisture from the surrounding

environment, critically alters particle size, shape, and refractive index (Burgos et al., 2019; Titos et al., 2021) and impacts the
angular distribution of scattered light and thus aerosol optical properties (Fierz-Schmidhauser et al., 2010; Zieger et al., 2015).



Since the globally measured long-term in situ aerosol measurements are standardized below 40 % relative humidity (RH)
(WMO/GAW, 2016), these "dry" conditions do not reflect the real atmosphere, leading to an inadequate understanding of
aerosol water uptake, which contributes to significant uncertainties affecting aerosol climate effects (Burgos et al., 2020; Myhre
et al., 2013; Ray et al., 2024).
The light scattering enhancement due to humidity can be expressed by the light scattering enhancement factor f(RH)
as in Eq. (1):
$$f(RH) = \frac{\sigma_{sp}(RH,\lambda)}{\sigma_{sp}(RH_{dry},\lambda)},\qquad\qquad\qquad\qquad\qquad\qquad\qquad\qquad\qquad\qquad\qquad (1)$$
where $\sigma_{sp}$ (RH,λ) and $\sigma_{sp}$ (RH$_{dry}$,λ) denote total scattering coefficients under elevated RH conditions and dry conditions
measured at the same wavelength λ, respectively (Covert et al., 1972). A similar formulation applies for backscattering,
f(RH)$_{bsp}$ (Titos et al., 2021).
Several approaches to investigate f(RH) have been proposed. Tandem-humidified nephelometer systems occurred in the
1960s and have undergone substantial innovations since then (Pilat and Charlson, 1966). These systems consist of one
nephelometer measuring under dry conditions and a second nephelometer measuring a humidified aerosol sample. Two main
instrumental set-ups were identified in the 26 tandem-humidified nephelometer measurements from ground-based sites
worldwide (Burgos et al., 2019). The "NOAA design" directs aerosol through a first dry and later humidified nephelometer
(e.g., Doherty, 2005; Liu and Li, 2018), while the "PSI design" splits the aerosol into parallel dry and humidified paths (e.g.,
Zieger et al., 2015, 2014). Both systems used an RH scanning regime (20 to 95 % RH) for the humified nephelometer (Titos
et al., 2016).
The comparison of the integrating nephelometer TSI 3563 with AURORA 3000 possessed an overall uncertainty of 2-5 %
for $\sigma_{sp}$ and 3-11 % for $\sigma_{bsp}$ in laboratory conditions, respectively (Müller et al., 2011). The experimental set-ups comprising
two or more instruments could introduce additional uncertainty to the resulting data, considering different sampling lines for
nephelometers, non-symmetrical apparatus, or the critical measurement part under highly humid conditions (Anderson et al.,

1996).

Thus, this study introduces a novel single-nephelometer system to reduce additional uncertainties in the f(RH) estimation
and investigates ambient aerosol particles' light scattering hygroscopic behavior at the suburban site. To the best of our
knowledge, only one study of aerosol hygroscopic behavior in a suburban/urban European environment was published (Titos
et al., 2014). This study provides a unique insight into light scattering enhancement in European urban/suburban environments.

## 2. Materials and Methods

### 2.1. Description of the site

The instrumentation set-up was developed and tested at the Institute of Chemical Process Fundamentals (ICPF) of the
Czech Academy of Sciences in Prague, Czech Republic. The ICPF also runs a Suchdol atmospheric station located on the
institute campus (50° 7′ 35″ N, 14° 23′ 5″ E, 277 m a.s.l., Figure 1). The station is a suburban site and an Aerosol In Situ
National Facility (AIS NF) of the ACTRIS ERIC (Aerosols, Clouds, and Trace gases Research InfraStructure, European
Research Infrastructure Consortium; https://www.actris.net/). The aerosol instruments are positioned within the sampling
container, with the sampling heads situated approximately 4 meters above the ground.
The station is located at the periphery of the plateau above the capital, Prague (1.37 million citizens in 2025), 5 km from
the city center. The site is surrounded by residential housing, utilizing gas as the primary energy source for heating (80 %),
while the remainder utilizes electricity (16 %), community heating (2 %), or burns solid and liquid fuels (less than 2 %) (Český
statistický úřad, 2021). The nearest road is situated at a distance of 250 m (10,000-15,000 cars per day, Vodička et al., 2013),
but no major road is located within 1 km of the site. The Václav Havel airport is situated 9 km SW of the site. The agricultural



fields are located within a 2 km radius to the west. The predominant wind direction at the site is from the WSW (mainly
summer and winter), with a notable influence of SE during winter and NW during spring (Fig. S.1). The measurement
campaign was conducted from 15 November 2022, to 19 August 2023.

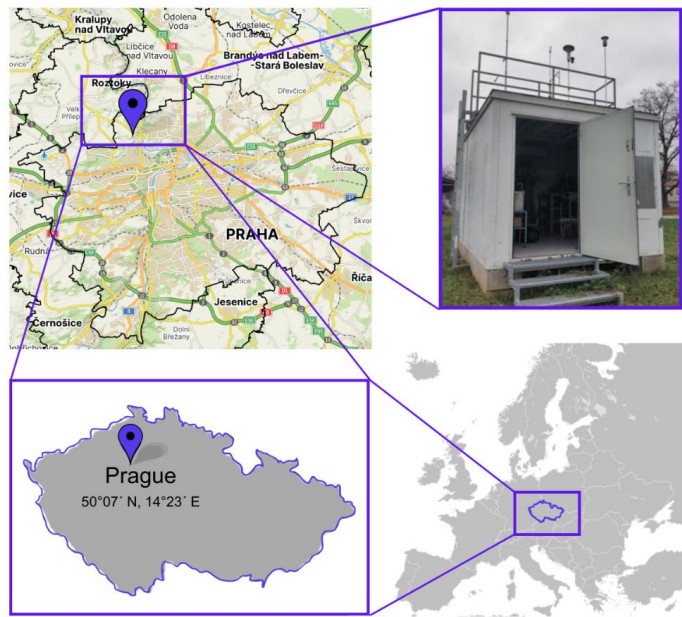


**Figure 1: Location of Suchdol ACTRIS ERIC AIS NF site within Prague, Czech Republic. Map source: © Seznam.cz, a.s.**
**2.2. The single-nephelometer instrumentation**
Aerosol particles were sampled through a $PM_{10}$ sampling head (Leckel, GmbH) and subsequently dried by a custom-built
Nafion dryer (Permapure) to achieve the RH level below 40 % (Figure 2, No. 1). The total aerosol flow of 10 lpm was divided
equally between two parallel sampling lines: a dry sampling line (5 lpm) and a humidified sampling line (5 lpm). The dry
sample was passed through the sampling system to the integrating nephelometer (TSI 3563) without additional adjustments.
The other line led to the second Nafion membrane, functioning as a water exchange medium, facilitating counter-current mass
transfer between humidified particle-free air and the aerosol sample (Figure 2, No. 3). This humidification process aimed to
achieve a sample RH $\geq$ 80 %.



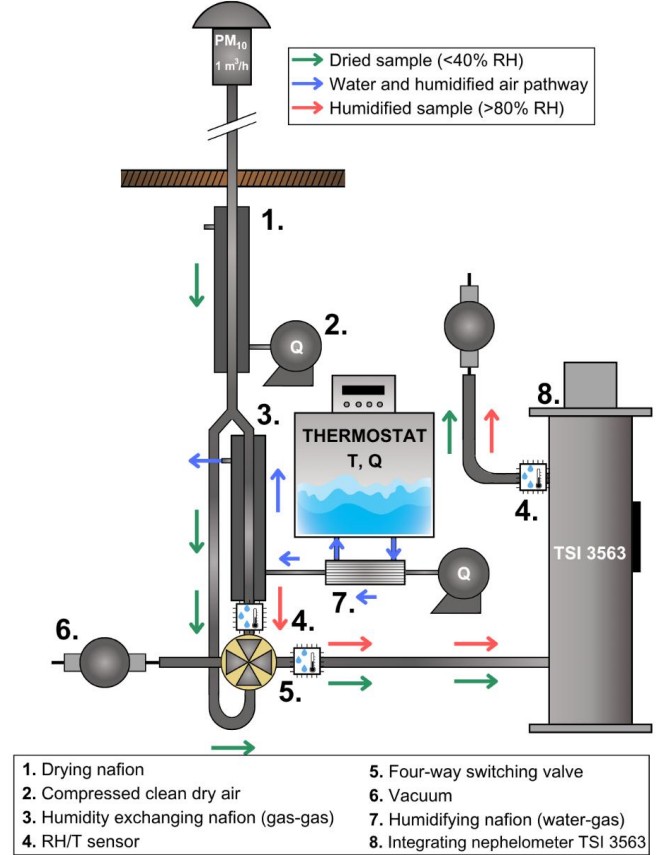


**Figure 2: A design of the single-nephelometer set-up system for studying aerosol hygroscopic behavior.**

Humid particle-free air was produced in the closed circulation system (Figure 2, blue arrows). Demineralized water,

heated in a controlled manner by the thermostat, was directed to the bundle Nafion membrane (Figure 2, No. 7), where mass
transfer between water (in channels) and the dry particle-free air (outside the channels) occurred. The excessive water was
later returned to the thermostat, and the humid particle-free air flowed into the humidity-exchanging Nafion membrane (Figure
2, No. 3). The temperature of the demineralized water and the flow rate of the humid particle-free air were regulated to achieve
the desired RH level in the humidity-exchanging Nafion. An RH/T sensor (HYT939, Innovative Sensor Technology, AG) was
installed before the switching valve to control the RH in the humidified sample. Since the RH sensor inside the measurement
cell was not sufficiently accurate, additional RH/T sensors were positioned immediately in front of the inlet and after the outlet
of the instrument (Figure 2, No. 4) to control the RH dynamics and to calculate the dew point temperature $T_{dew}$, which was
used to estimate the real RH of the sample (see Chapter 2.4 Data treatment). All RH/T sensors were calibrated against a
standard thermometer (F250 MKII, Automatic System Laboratories) and a dew point mirror (CMH2, Alpha Moisture Systems)
at the beginning of the campaign.

Every 60 minutes, the four-way switching valve (Figure 2, No. 5) automatically directed either the dry or the

humidified sample to the TSI 3563 integrating nephelometer (Figure 2, No. 8). First, a 10-minute conditioning period was
initiated to reach the target RH of the sample, followed by a 50-minute measurement period as determined and optimized by
pilot testing. During the dry sample measurement, the automatic switching valve allowed the dry sample to flow directly to



the nephelometer, and the humidified sample was directed to the exhaust (Figure 2, No. 6). And vice versa, when the humidified
aerosol sample was sampled to the nephelometer, the dry sample was discarded.

The thermostat temperature and the flow rate of the humidified particle-free air were checked regularly to ensure

proper humidification of the sample and to prevent condensation inside the instrument. All parts sensitive to changes in RH
(humidity-exchanging Nafion, tubing within the closed humidity circuit, and inlet tubing to the nephelometer) were insulated
to prevent heat losses and water condensation.

Upon reaching the nephelometer measuring cell, a dry or humidified aerosol sample was illuminated with a halogen

lamp at an angle range of 7°–170°. The scattered light passed through three band-pass filters and was detected in
photomultiplier tubes (PMT) at 450, 550, and 700 nm wavelengths. The resulting total scattering and backscattering
coefficients ($\sigma_{sp}$ and $\sigma_{bsp}$) with a time resolution of 1 minute.
The nephelometer was calibrated twice a day with particle-free air and fully calibrated every 2–3 months with $CO_2$ as the high-
span gas and particle-free air as the low-span gas, always in the dry measurement regime. The continuous dry measurement
was performed approximately once a month (overnight) to avoid water condensation inside the instrument.

However, it should be stated that such a measurement approach possesses the limitation of reduced time resolution and a

lack of parallel measurement of dry and wet aerosol properties, important for humidogram analyses.
**2.3. Auxiliary measurements**

The Mobility Particle Size Spectrometer (MPSS) measured the aerosol particle number concentration with a time

resolution of 5 minutes, using a custom-built Differential Mobility Analyzer (DMA, both TROPOS, Germany), positive high-
voltage power supply, and Condensational Particle Counter (CPC 3772, TSI). The MPSS ranged from 10 to 800 nm, 32 size
channels per decade, with data further subdivided into size modes: 8–100 nm, 100–200 nm, 200–500 nm, and particles above
500 nm for the analysis. An additional total count CPC (3750, TSI) measured the total particle number concentration of
particles larger with $d_p(50)$ at 10 nm.

Elemental and organic carbon (EC and OC) concentrations were measured from November 2022 to July 2023 using

a semi-online field analyzer from Sunset Laboratory Inc. (USA) (Bauer et al., 2009). The analyzer was connected to a $PM_1$
(November – December 2022) and $PM_{2.5}$ (rest of the period) inlet with a flow rate of 8 lpm. Samples were collected at 2-hour
intervals on a quartz fiber filter and analyzed according to the shortened EUSAAR2 protocol (Cavalli et al., 2010). Each
measurement was corrected for charring, and the RTCalc726 software automatically determined the split point between EC
and OC using a linear fit of laser and temperature corrections. OC was divided into fractions based on temperature. The most
volatile fractions, OC1 and OC2, volatilize at 200 °C and 300 °C and are commonly present in fresh vehicle exhaust, biomass
burning, and coal combustion (Shen et al., 2025; Vodička et al., 2015). OC3 and OC4 subfractions volatilize at 450 °C and 650
°C and represent less volatile fractions of OC with higher molecular weights and are associated with chemical aging and the
products of photochemical reactions (Aswini et al., 2019; Shen et al., 2025). The instrument was equipped with a parallel
carbon plate denuder to eliminate volatile organic compounds and prevent positive bias in OC measurements. Instrument
blanks were recorded daily at midnight.

A multiwavelength aethalometer (Model AE33, Magee Scientific, USA, 2018) continuously measured light

absorption by particles at seven wavelengths (370, 470, 520, 590, 660, 880, and 950 nm). Particles were sampled through the
$PM_{10}$ sampling head (Leckel GmbH) at a flow rate of 5 lpm, dried in a custom-made Nafion dryer (TROPOS, Leipzig,
Germany), and deposited onto tetrafluoroethylene (TFE) coated glass filter tape. Light transmission through the deposited
sample is measured and compared to the blank filter tape spot as a reference, resulting in absorption coefficient ($\sigma_{ap}$, Mm$^{-1}$)
and equivalent black carbon concentration (eBC, µg m$^{-3}$) data.





Ambient temperature (T), RH, wind speed (WS), wind direction (WD), global solar radiation (GLRD), and ozone (O3)
concentration were measured hourly in the Czech Hydrometeorological Institute container located next to the ACTRIS AIS
container with all the aerosol instruments.
**2.4. Data treatment**
**2.4.1. Humidified nephelometer system**
Four subsystems working simultaneously were needed to obtain valid datasets from the measurements: nephelometer
measurement, automatic switching between dry and humid measurements, humidification of the sample, and RH/T sensors.
The TSI 3563 integrating nephelometer data set was processed according to the EMEP Standard Operating Procedure.
The raw $\sigma_{sp}$ and $\sigma_{bsp}$ data at all wavelengths were validated (removal of invalid, missing, and calibration data). Values below
the limit of detection (LOD) were replaced by LOD/2 values, corrected to a non-Lambertian illumination according to
Anderson and Ogren (1998), and standardized to STP conditions (273.15 K, 1013.25 hPa).
The automatic four-way switching valve was controlled using a custom-made LabVIEW program. The resulting data
set was paired with the nephelometer data set to identify dry and humidified measurements.
All datasets produced by the RH/T sensors regulating the thermostat temperature and the flow of humidified particle-free air
to achieve the desired RH of the sample were also recorded using a custom-made LabVIEW program.
Temperature and RH data from all three sensors were corrected using calibration curves derived from comparing with
the referenced thermometer and the dew point mirror. The real RH of the sample was derived by assuming that the dew point
temperatures in front of and behind the cell are similar (Ren et al., 2021). The approximated Magnus-Tetens formula (Alduchov
and Eskridge, 1997) was used to calculate the dew point temperatures of both RH/T and from the mean $T_{dew,}$ the saturation
vapor pressure, $p_{sat}$, at $T_{dew}$ was calculated as in Eq. (2):
$$p_{sat} = a * e^{\frac{b*T_{dew}}{T_{dew}+c}},$$    (2)
where a, b, and c are empirical constants derived from the experimental data: a = 6.112 hPa, b = 17.67, and c = 243.51 (b and
c are dimensionless); $T_{dew}$ in °C.
A similar approach was used to calculate the saturation vapor pressure at the temperature in the measuring cell, $T_{cell}$ — $p_{cell}$.
The final RH of the sample in the measuring cell, $RH_{cell}$, was calculated as follows in Eq. (3):
$$RH_{cell} = \left(\frac{p_{sat}}{p_{cell}}\right) * 100.$$    (3)
The $RH_{cell}$ data set was later combined with the nephelometer and switching valve data. $RH_{cell}$ with valve position variables
was used to separate the data into dry (RH ≤ 40 %) and humidified (RH ≥ 80 %) datasets, with data corresponding to the range
of 40 % < RH < 80 % being discarded. The dry and humidified datasets were averaged every hour.
The enhancement factors f(RH) and f(RH)$_{bsp}$ were calculated to obtain information about light scattering enhancement
due to hygroscopicity. In this study, f(RH) was calculated based on "humid-centered" and "dry-centered" intervals to avoid the
influence of possible extreme pollution events at the site (Fig. 3). The humid-centered interval was calculated by dividing the
average humidified $\sigma_{sp}$ value by the mean of two lateral 1-hour averaged dry $\sigma_{sp}$ (Fig. 3, gray). The dry-centered interval was
calculated by dividing the average of two lateral 1-hour averaged humidified $\sigma_{sp}$ by the 1-hour averaged dry $\sigma_{sp}$ value (Fig. 3,
ochre). Extreme and invalid values were inspected and discarded if necessary. The data coverage of the entire measurement
campaign, including individual seasons, as well as the variability of RH and temperature under humid and dry conditions, can
be found in Table S.1, Table S.2, and Fig. S.1 in Supplementary Materials.



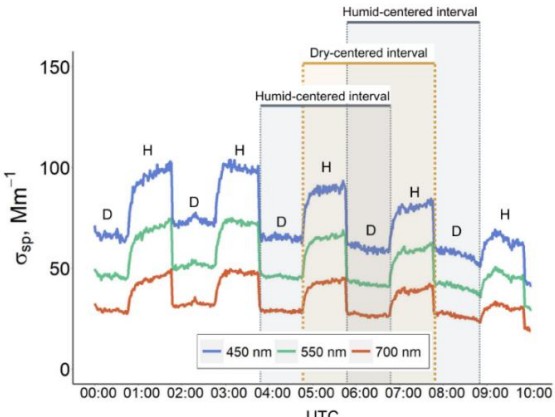


**Figure 3: The example of f(RH) calculation from a single-nephelometer measurement on December 12, 2022. The D and H symbols**

**indicate "dry" and "humid" measuring intervals of $\sigma_{sp}$.**
**2.4.2. Relationship between aerosol scattering and absorption**
To describe the spectral dependence of light scattering, the Scattering Ångström Exponent SAE was calculated as
(Clarke and Kapustin, 2010):
$$\text{SAE}_{\lambda_1-\lambda_2} = -\frac{\log\left(\frac{\sigma_{sp}(\lambda_1)}{\sigma_{sp}(\lambda_2)}\right)}{\log\left(\frac{\lambda_1}{\lambda_2}\right)},$$   (4)
where $\lambda_1$ and $\lambda_2$ are the wavelengths of light at which $\sigma_{sp}$ was measured. SAE contains information about aerosol size: SAE
values < 1 indicate the predominance of particles in the coarse mode, while SAE values ≥ 2 indicate a predominance of the
aerosol fine mode (Seinfeld and Pandis, 2006).
In addition to SAE, ΔSAE was defined (Perrone et al., 2018) in Eq. (5):
$$\Delta\text{SAE} = \text{SAE}_{450-550} - \text{SAE}_{550-700},$$   (5)
and provides insight into the relative contribution of fine and coarse mode particles and whether the particle size distribution
is mono-, bi-, or multimodal. Positive ΔSAE values indicate the presence of two distinct modes—a fine mode and a coarser
one—while negative ΔSAE values suggest the dominance of a single fine particle mode (Perrone et al., 2018).
The Absorption Ångström Exponent (AAE), which describes a spectral dependence of light absorption, was
calculated analogously from $\sigma_{ap}$ (Mbengue et al., 2021). AAE can provide information on chemical composition: AAE values
< 1 could indicate BC core or non-absorbing coating particles, AAE values around 1 are classified as BC aerosol, while AAE
values around 2 and higher indicate light absorption in ultraviolet and blue spectral regions, suggesting the presence of organic
carbon — brown carbon BrC in this study — or mineral dust (Cappa et al., 2016).
To estimate the dominant aerosol type at the site, we used the AAE vs. SAE plot from Cappa et al. (2016), which can
estimate the potential aerosol type without direct information on the chemical composition. For the aerosol type assessment in
this study, the AAE was calculated at 520–660 nm and the SAE at 450–550 nm. The AAE vs. SAE plots were additionally
color-coded with the Single Scattering Albedo (SSA), the ratio of the aerosol light scattering, and the total aerosol light
extinction (light scattering plus absorption) at the predefined wavelength λ defined in Eq. (6):
$$SSA_\lambda = \frac{\sigma_{sp}(\lambda)}{\sigma_{sp}(\lambda)+\sigma_{ap}(\lambda)}.$$   (6)



Sites predominantly influenced by aerosol scattering (clean marine or remote Arctic sites) exhibit SSA values close to 1, while
anthropogenically influenced sites exhibit significantly lower SSA (Pandolfi et al., 2018). In this study, also the humidified
equivalent of SSA was calculated from dry measurements of $\sigma_{ap}$ (RH < 40 %) and humidified $\sigma_{sp}$ (RH > 80 %).
The asymmetry factor $g$ describes the angular distribution of the scattered light and is defined as the average cosine
of the angle between the incident light and the scattered beam θ, weighted by the probability of scattering for each possible
angle. Based on the Henyey-Greenstein approximation (Andrews et al., 2006; Wiscombie and Grams, 1976):
$g_\lambda = -7.143889b_\lambda^3 + 7.464439b_\lambda^2 - 3.96356b_\lambda + 0.9893,$  (7)
where $b$ is the hemispheric backscattering ratio. $g$ ranged from -1 for completely back-scattered light to 1 for completely
forward-scattered light and is one of the essential inputs for the radiative transfer models.
The hemispheric backscattering ratio $b$ denotes the fraction of light scattered back to the upper hemisphere of the
particle and the total scattered light and can be measured directly from the optical instrument without knowledge of the
scattering phase function (ranging from 0 to 1) calculated following Eq. (8):
$b_\lambda = \frac{\sigma_{bsp}(\lambda)}{\sigma_{sp}(\lambda)}$  (8)
$g$ and $b$ are particularly useful for distinguishing aerosol types and assessing their radiative impacts, as backscattering plays a
critical role in determining the cooling efficiency of atmospheric particles.

### 2.4.3. Back trajectory analysis

The dry optical properties data were paired with a cluster analysis of back trajectories calculated using the
HYSPLIT_4 model from the NOAA Air Resources Laboratory to understand the sources of distinct aerosol types better. The
global data assimilation system (GDAS) at 1° x 1° resolution (Draxler and Hess, 1998; Stein et al., 2015) was used as
meteorology input, and 72-hour air mass back trajectories arriving at 200 m a.g.l. were calculated every 6 hours. The number
of clusters was estimated based on total spatial variance.

### 2.4.4. New particle formation events

New Particle Formation (NPF) days were estimated using MPSS Level 1 data (https://ebas-
submit.nilu.no/templates/Differential-Scanning-Mobility-Particle-Sizer/lev1, last accessed: April 2025). Daily heat maps of
concentrations, mode positions, and concentration of particles < 20 nm and between 20 and 100 nm were plotted for every
day, following the Dal Maso et al. (2005) method altered by Németh et al. (2018) for urban stations, each day of the observed
period was classified as either NPF event, non-event, undefined day or missing day.
The missing day class (16 out of 304) was assigned when more than 6 hours of the daily measurements were missing,
disabling the classification. Non-event days (67 out of 304) were those where a mode below 25 nm occurred only for a shorter
time than one hour or was absent. Undefined days were identified on 123 days, as the local traffic emissions were often
indistinguishable from potential NPF. NPF days (105 out of 304) were defined as days with a new mode below 25 nm occurring
during the day and with an observable particle size growth for more than 1 hour.

### 2.4.5. The Principal Component Analysis

To investigate the relationship and underlying patterns between f(RH), f(RH)$_{bsp}$, aerosol particle number
concentration in different modes, meteorological variables, and NPF events, a Principal Component Analysis (PCA) was
performed. The PCA was carried out in R (version 2023.12.0) using the PCA function from the FactoMineR package to reduce
the data's dimensionality and identify dominant modes of variability in the dataset.





Prior to PCA, all variables were standardized (zero mean, unit variance). To handle the remaining missing data without biasing
the results, we used a PCA-based imputation method (missMDA package). The optimal number of dimensions for imputation
was first estimated using the estim_ncpPCA function; missing values were then imputed using the imputePCA function, which
reconstructs incomplete observations based on the relationships captured by the principal components. The first few principal
components (Dims) explaining the largest proportion of variance were retained for interpretation based on scree plot analysis
and cumulative variance thresholds (Smith, 2002).

## 3. Results and discussion

### 3.1. Light scattering properties

The overall median value of dry $\sigma_{sp}$ at 550 nm (text refers to the overall measurement at $\lambda = 550$ nm unless stated otherwise)
was 28.45 Mm$^{-1}$ at the studied site, corresponding to the range of values observed at urban and suburban sites: 14.83 Mm$^{-1}$ at
SIRTA (FR), 47.39 Mm$^{-1}$ in Athens (GR), 39.83 Mm$^{-1}$ in Lecce (IT), 18.04 Mm$^{-1}$ and 43.14 Mm$^{-1}$ in Madrid and Granada (ES)
(Donateo et al., 2020; Pandolfi et al., 2018).
The dry SAE$_{450-700}$ value of 1.65, together with a positive median dry ΔSAE (Table 1), indicated a predominantly
fine-mode aerosol population, with spectral curvature suggesting the presence of a secondary mode associated with larger
particles, likely from aging or mixing processes, similarly to Athens and Granada (SAE$_{450-700}$ of 1.6 and 1.69), while in Lecce,
the fine particle mode dominated (SAE$_{450-700}$ of 1.84) (Donateo et al., 2020; Pandolfi et al., 2018). The dry $b$ (0.161) and $g$
(0.521) were also typical for urban/suburban environments, suggesting a slightly stronger cooling potential of the aerosol
population compared to other sites.
Table 1: The statistics of light scattering properties in the PM$_{10}$ fraction at different wavelengths. P25, P75, and P50 denote the 25$^{th}$
and 75$^{th}$ percentiles and median, respectively. All variables except f(RH) and f(RH)$_{bsp}$ were measured and calculated at RH < 40 %,
and all variables except $\sigma_{sp}$ and $\sigma_{bsp}$ are dimensionless.

| | | | | Whole period | | Fall | Winter | Spring | Summer |
|---|---|---|---|---|---|---|---|---|---|
| | $\lambda$ | P25 | P50 | P75 | mean ± SD | | | P50 | |
| | 450 nm | 1.19 | 1.30 | 1.42 | 1.33 ± 0.34 | 1.20 | 1.27 | 1.34 | 1.33 |
| f(RH) | 550 nm | 1.20 | 1.32 | 1.44 | 1.35 ± 0.37 | 1.22 | 1.29 | 1.36 | 1.34 |
| | 700 nm | 1.31 | 1.57 | 1.93 | 1.66 ± 0.54 | 1.46 | 1.54 | 1.61 | 1.63 |
| | 450 nm | 1.07 | 1.22 | 1.41 | 1.25 ± 0.32 | 1.20 | 1.20 | 1.25 | 1.20 |
| f(RH)$_{bsp}$ | 550 nm | 1.04 | 1.12 | 1.20 | 1.14 ± 0.25 | 1.08 | 1.10 | 1.16 | 1.12 |
| | 700 nm | 1.11 | 1.22 | 1.34 | 1.25 ± 0.32 | 1.15 | 1.20 | 1.27 | 1.22 |
| | 450 nm | 21.81 | 40.31 | 80.29 | 60.99 ± 58.55 | 99.66 | 57.14 | 38.99 | 33.10 |
| $\sigma_{sp}$ (Mm$^{-1}$) | 550 nm | 15.54 | 28.45 | 56.29 | 44.19 ± 43.50 | 73.22 | 40.84 | 27.99 | 22.96 |
| | 700 nm | 10.89 | 19.27 | 36.76 | 29.93 ± 29.47 | 48.69 | 27.27 | 19.20 | 15.33 |
| | 450 nm | 3.37 | 6.05 | 10.33 | 8.14 ± 7.27 | 12.05 | 7.48 | 5.74 | 5.15 |
| $\sigma_{bsp}$ (Mm$^{-1}$) | 550 nm | 2.67 | 4.72 | 8.13 | 6.42 ± 5.70 | 9.54 | 5.86 | 4.56 | 3.97 |
| | 700 nm | 2.29 | 4.08 | 7.13 | 5.62 ± 5.04 | 8.61 | 5.23 | 3.95 | 3.34 |
| SAE$_{450-700}$ | | 1.45 | 1.65 | 1.85 | 1.61 ± 0.34 | 1.56 | 1.57 | 1.70 | 1.81 |
| AAE$_{470-660}$ | | 1.30 | 1.47 | 1.62 | 1.47 ± 0.36 | 1.58 | 1.56 | 1.47 | 1.26 |
| ΔSAE | | -0.091 | 0.008 | 0.194 | 0.063 ± 0.183 | -0.122 | -0.024 | 0.056 | 0.100 |
| SSA | 550 nm | 0.636 | 0.709 | 0.766 | 0.700 ± 0.095 | 0.650 | 0.672 | 0.709 | 0.755 |
| $b$ | | 0.141 | 0.161 | 0.174 | 0.158 ± 0.027 | 0.134 | 0.151 | 0.163 | 0.171 |
| $g$ | | 0.492 | 0.521 | 0.562 | 0.528 ± 0.106 | 0.578 | 0.542 | 0.516 | 0.498 |

However, a relatively low median value of SSA$_{550}$ (0.709) implies a substantial influence of absorbing aerosol species in
the aerosol population. The median AAE$_{470-660}$ value of 1.47 suggests a relatively balanced contribution of black carbon (BC)
and brown carbon (BrC) to aerosol absorption, with a stronger influence of BC in summer (AAE$_{470-660}$ of 1.26), likely due to



traffic emissions. In contrast, elevated AAE values in fall (1.58) and winter (1.56) suggest an increased contribution of BrC
from biomass burning, possibly related to residential heating.

### 3.2.  f(RH) and f(RH)$_{bsp}$

The overall low enhancement factors f(RH) and f(RH)$_{bsp}$ of 1.32 and 1.12 suggest the influence of low-hygroscopic
carbonaceous aerosol species from local combustion, traffic sources, and their aged derivatives. This result is consistent with
the lower range of f(RH) values observed at urban and suburban sites, for example, average values ranging between 1.32 and
1.74 in a suburban area of Beijing in autumn (Ren et al., 2021), 1.30 near Manacapuru city in Brazil, a site influenced by
industrial activities with soot, high-sulfur oil emissions and biomass burning (Burgos et al., 2019), or $1.5 \pm 0.2$ in winter in
Granada (Titos et al., 2014).

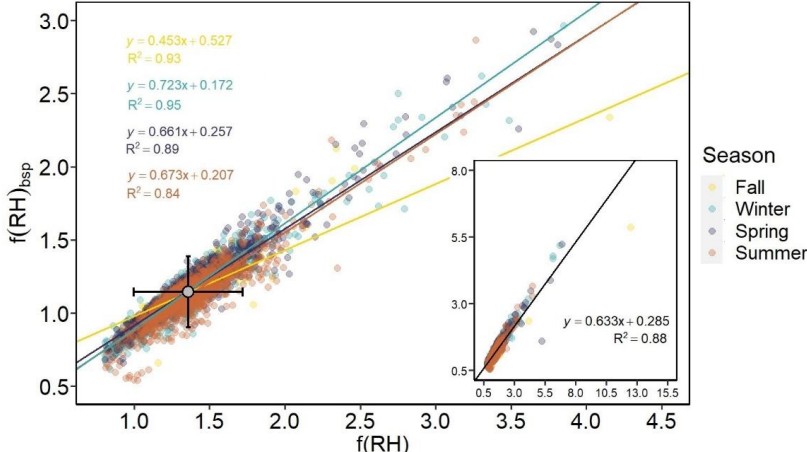

**Figure 4: The weighted bivariate fit of f(RH)$_{bsp}$ vs the f(RH) at $\lambda$ = 550 nm in individual seasons. The grey point represents the overall**
**f(RH) and f(RH)$_{bsp}$ mean value with error bars. The inset demonstrates the weighted bivariate fit for the whole dataset.**

The f(RH)$_{bsp}$ mimicked the behavior of the f(RH) during the whole year, with the highest correlation in winter (Figure
4). Compared to Titos et al. (2021), our median f(RH) and f(RH)$_{bsp}$ values (gray point in Figure 4) fall into the low enhancement
category of urban sites. Despite the strong linear relationship, f(RH)$_{bsp}$ does not precisely mirror f(RH) and varies with seasons.
As aerosol ERF depends on the hemispheric backscattering ratio *b*, models relying solely on assumed f(RH) can lead to
inaccurate outcomes (Haywood and Shine, 1995; Hegg et al., 1996; Titos et al., 2021). Such regressions can be beneficial
given the scarcity of f(RH)$_{bsp}$ measurements for modeling enhancement for specific site types. Titos et al. (2021) performed
this fit across several environments (Arctic, marine, rural, and urban) and found the fit equation of y = 0.55x + 0.32, $R^2$=0.69
for the Melpitz urban site. Parameters retrieved from our weighted bivariate fit of f(RH)$_{bsp}$ vs the f(RH) can be found in Table
S.3.
The f(RH) depends on RH, the particle size, chemical composition, and light wavelength. Its spectral dependence is
crucial for radiative forcing estimates (Fierz-Schmidhauser et al., 2010; Kiehl and Briegleb, 1993; Titos et al., 2021). At the
studied site, the f(RH) increases with wavelength in most cases and seasons except summer (frequency distributions centered
around 0, Fig. S.3), aligning well with results from urban sites (Titos et al., 2021) but showing lower values compared to
marine or Arctic environments. Occasionally, an opposite behavior with a decrease f(RH) with increasing wavelength was
observed, linked to dust episodes and particle size shift (Carrico et al., 2003; Fierz-Schmidhauser et al., 2010).
Based on Mie's theory, Hegg et al. (1996) proposed that f(RH)$_{bsp}$ should be approximately 25 % lower than f(RH) for
typical atmospheric aerosols. Our observations support this, showing consistently smaller f(RH)$_{bsp}$ than f(RH) across all





wavelengths. On average, f(RH)$_{bsp}$ was lower by 6 %, 15 %, and 22 % at 450, 550, and 700 nm, respectively, with the largest
differences observed in summer reaching 10 %, 16 %, and 25 %. The relative difference between f(RH) and f(RH)$_{bsp}$ increases
with wavelength, suggesting a spectral sensitivity of backscattering to humidification, likely due to particle size and
composition effects on the angular distribution of scattered light. These findings highlight the need for wavelength- and season-
specific correction factors when using f(RH) to estimate aerosol backscattering or when interpreting satellite data sensitive to
the backscattered light.
The probability density functions of f(RH) and f(RH)$_{bsp}$ for different wavelengths can be found in Fig. S.4.

### 3.3. Seasonal variability

The lowest f(RH) and f(RH)$_{bsp}$ values were observed in the fall, with a monthly minimum in November (1.22 and 1.07,
Figure 5). Both peaked in spring, with a monthly maximum in May (1.40 and 1.19). The increase from autumn to spring was
interrupted in March when both f(RH) and f(RH)$_{bsp}$ dropped to 1.22 and 1.08, probably due to March being a transitional
period between winter and spring in the Northern Hemisphere. Although this anomaly has not been fully understood yet,
CAMS reanalysis by Flemming et al. (2017) also identified irregular springtime atmospheric patterns over the Northern
Hemisphere, while Suchánková et al. (2025) observed a steep increase in dry $\sigma_{sp}$ and $\sigma_{bsp}$, and ultrafine and fine particle number
concentration at an urban site in France. After peaking in May, both f(RH) and f(RH)$_{bsp}$ gradually decreased towards August
(1.34 and 1.12). The seasonal variation is consistent with results reported at the urban site in Granada (Titos et al., 2014).
Lower values of f(RH) and f(RH)$_{bsp}$ in fall and winter suggest the prevalence of low-hygroscopic aerosol species,
such as carbonaceous particles from combustion and traffic sources. In summer and spring, the enhancement is influenced by
hygroscopic SOA, as supported by the statistically significant Spearman correlations between f(RH)/f(RH)$_{bsp}$ and SSA$_{550}$ (R
= 0.31/0.23). Although SSA$_{550}$ peaks in summer, likely due to highly hygroscopic secondary organic and inorganic aerosols
(sulfates and nitrates), the strongest correlation between f(RH)/f(RH)$_{bsp}$ and SSA$_{550}$ is in spring (R = 0.34/0.27). Sulfates and
nitrates amplify aerosol hygroscopicity more effectively than organic matter, especially compared to the less hygroscopic
organic species prominent in summer. This seasonal contrast highlights the key role of inorganic species like sulfate and nitrate
in controlling aerosol optical properties through their impact on hygroscopic growth. During spring, the aerosol composition
is relatively homogeneous, dominated by aged, regionally transported aerosols with intermediate hygroscopicity, resulting in
stronger and more predictable correlations between enhancement factors and SSA$_{550}$. While both spring and summer can
include biogenic organics and aged urban particles, the chemical composition in summer is more variable, leading to less
consistent hygroscopic behavior and the weakest seasonal correlations with SSA$_{550}$ (R = 0.16/0.17).

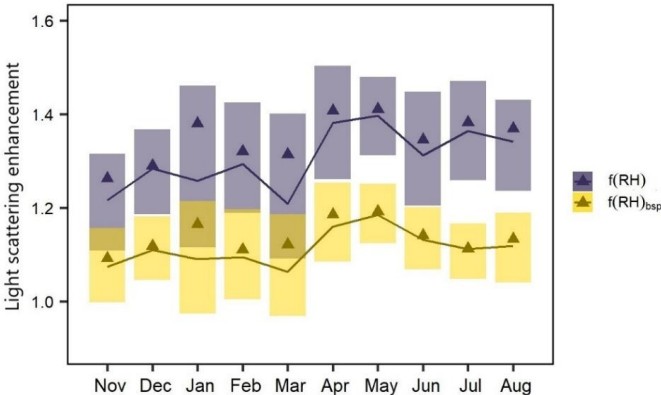

**Figure 5: The monthly variation of the f(RH) and the f(RH)$_{bsp}$ measured at 550 nm. The lines with shaded areas depict the median**
**with the interquartile range, and the triangles represent mean values.**



The temporal variation of f(RH) did not show any consistent connection to SAE$_{450-700}$ (Fig. S.5a). The SAE$_{450-700}$ values
rise from colder months to warmer months, suggesting the presence of smaller particles from the NPF or the traffic. Compared
to winter, the enhanced mixing and higher dilution due to the higher planetary boundary layer height (PBLH) in spring and
summer also prevent particle growth due to condensation. This statement was supported by the analysis of particle number
size distribution (PNSD), with D$_p$ below 200 nm primarily present in the photooxidatively active time of the year. At the same
time, larger particles occurred mainly in colder seasons (Fig. S.6). However, the nephelometer does not precisely measure the
ultrafine particle sizes due to its geometry and principle of operation, possibly distorting the direct relationship between the
light scattering enhancement and PNSD.
While we found a larger influence of the chemical composition on light scattering enhancement than the particle size, the
seasonal variations in the ratio *b* were linked to changes in aerosol size distributions and sources. Smaller particles (higher
SAE$_{450-700}$ values in warmer months) typically exhibit a higher backscattering fraction under dry conditions; however, their
hygroscopic growth may lead to a relative decrease in backscattering enhancement compared to total scattering due to changes
in the scattering phase function (Fig. S.5b).

**3.4.  Light scattering enhancement vs other aerosol-intensive optical properties**

Understanding the relationship between light scattering enhancement and other aerosol optical properties can help improve
the characterization of aerosol types and their radiative impacts.
SAE$_{450-700}$ > 1 mirrored the frequency distribution of both f(RH) and f(RH)$_{bsp}$ (Figure 6). A slight shift towards more
hygroscopic behavior was observed for SAE$_{450-700}$ < 1 in f(RH)$_{bsp}$. In contrast, SSA$_{550}$ < 0.6 exhibits a shift to the left for both
f(RH) and f(RH)$_{bsp}$, indicating an overall significant decreasing effect of absorbing aerosol species on aerosol light scattering
enhancement, consistent with the finding of Titos et al. (2014).
In winter and spring, particles with SAE$_{450-700}$ < 1 showed slightly more hygroscopic behavior both for f(RH) and
f(RH)$_{bsp}$, possibly due to the internal mixtures of sulfates and nitrates or the presence of mineral dust. Particles characterized
with SSA$_{550}$ < 0.6 consistently showed lower f(RH) and f(RH)$_{bsp}$ values. In spring and summer, the frequency distribution of
SSA$_{550}$ > 0.6 mimicked the whole dataset distribution, implying a higher share of light-scattering aerosols (Fig. S.7).

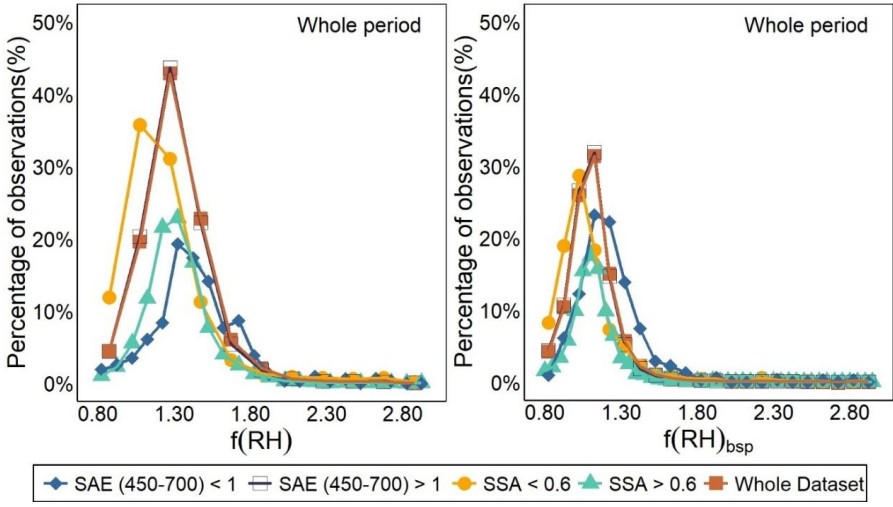


**Figure 6: Frequency distribution of f(RH) (left) and f(RH)$_{bsp}$ (right) calculated at 550 nm during the whole measurement campaign.
The distributions are color and symbol-coded based on SAE$_{450-700}$ and SSA$_{550}$ values.**





Dry $SSA_{550}$ was lower for less hygroscopic particles (f(RH) < 1.5) than for more hygroscopic ones (f(RH) > 1.5),
0.708 vs 0.742 (Fig. S.8a), confirming the adverse effect of absorbing components on hygroscopicity. Across all seasons, the
SSA increased with RH (Figure 7b), while $b_{550}$ showed the opposite behavior (Figure 7a), as expected, given their inverse
relationship. This relationship was more pronounced under humidified conditions than dried ones (Fig. S.8b) (Burgos et al.,
2020; Titos et al., 2021). As particles take up water and grow, the total scattering increases with a predominance of the forward
light scattering over backscattering.

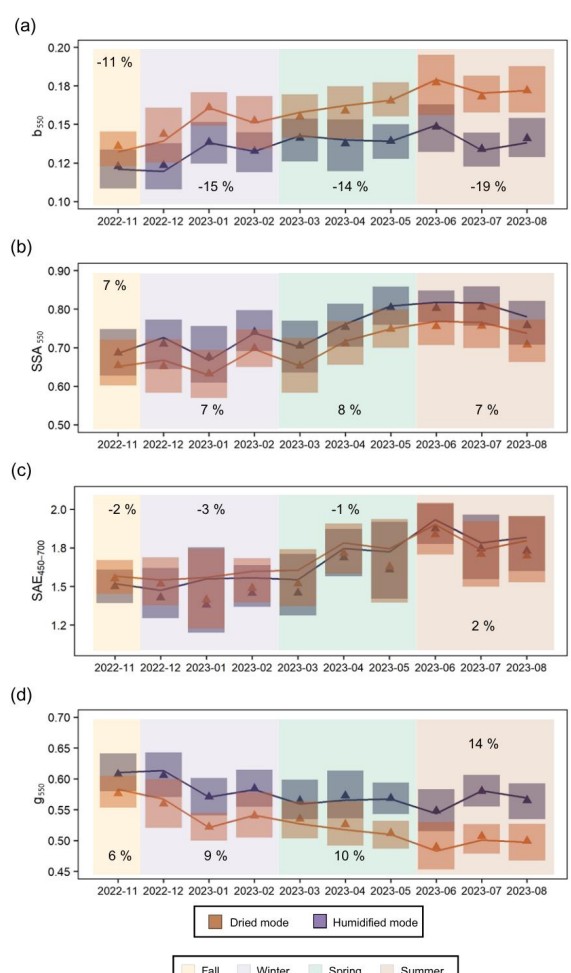


**Figure 7: The monthly variation of $b_{550}$ (a), $SSA_{550}$ (b), $SAE_{450-700}$ (c), and $g_{550}$ (d) at dried and humidified conditions in individual seasons (colored background). A percentage value in each season shows a relative change between humidified and dried median values for the respective season. The lines with shaded areas depict the median and the interquartile range, and the triangles represent respective mean values.**

The asymmetry factor $g_{550}$ also increased with RH, confirming stronger forward than backward scattering with enhanced
RH (Figure 7d). The $SAE_{450-700}$ decreased with increasing RH, indicating particle size growth in all seasons except summer
(Figure 7c). During summer, an atypical behavior of the $SAE_{450-700}$ was observed, with dry particles appearing optically larger
(lower SAE) than the humidified ones. This is likely linked to the dominance of BC-rich aerosols, which are weakly
hygroscopic and maintain large agglomerated structures in the dry state. Such structures may compact upon humidification,



reducing optical growth and increasing SAE. This behavior contrasts with highly hygroscopic sulfate- or nitrate-rich aerosol regimes, where humidification leads to particle growth and lower SAE values.

Comparing our site with those reported by Titos et al. (2021), we found the lowest f(RH), similar to urban (e.g., Manacapuru, Brazil, and Nainital, India), desert (Niamey, Niger), and some rural (e.g., Hyytiälä, Finland) environments. Interestingly, the SSA$_{550}$ at our site was the lowest, indicating a strong presence of absorbing aerosols such as black carbon or other combustion-related species (Fig. S.5c). This confirms the role of local or regional combustion sources in shaping aerosol optical properties.

### 3.5. The estimation of chemical composition from optical properties

Due to the lack of direct measurement of chemical composition at the site, we used the approach introduced by Cappa et al. (2016), using AAE$_{520-660}$, SAE$_{450-550}$, and SSA$_{550}$ to estimate the aerosol chemical composition. Aerosol at the station was found mainly in the fine fraction, dominated by BC in a mixture with BrC (Figure 8), in agreement with the low f(RH) and f(RH)$_{bsp}$ at the site. The site was also influenced by BC and BrC in a mixture with dust, probably coming from the road dust, although unexamined Saharan dust episodes may have contributed. These mixtures included large, poorly absorbing coated particles and locally emitted sulfates and nitrates in the fine size mode, which highly scatter but weakly absorb the incoming light.

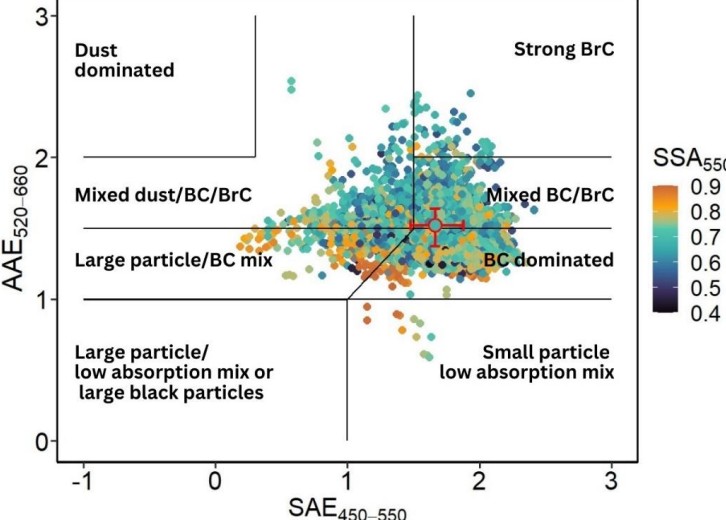

**Figure 8: AAE$_{520-660}$ vs. SAE$_{450-550}$ hourly means from the whole campaign, color-coded with SSA$_{550}$ overlaid with the aerosol characterization matrix adopted from Cappa et al. (2016) and Cazorla et al. (2013). The red circle with red error bars estimated the median AAE$_{520-660}$–SAE$_{450-550}$ point with interquartile ranges.**

SSA$_{550}$ values were equally distributed among the plot regions, except for the highest values in the aerosol mixture of sea salt, dust, and low-absorbing coated particles with BC and BrC. Such aerosol was typically connected with air masses coming from Poland and Eastern Europe (Figure **9**). High concentrations of the mixture with BrC were observed in fall, winter, and spring, predominantly due to biomass burning. Consequently, the median AAE$_{520-660}$ vs. SAE$_{450-550}$ point fell into the Mixed/BC/BrC" plot region. In contrast, the BC aerosol mixture started to dominate in spring and summer (Fig. S.9).

Although multiple wintertime clusters originated from the marine regions, only Cluster 6 (Figure 9a) shifted towards dust/large low-absorbing particle aerosol mixture, indicating the influence of fossil fuel/biomass burning sources during this



season. In spring, the air masses of Clusters 2 and 4 originated from the Norwegian Sea and Atlantic Ocean (Figure 9b) and
featured hygroscopic sea salt mixtures. In summer, the BC-dominated aerosol prevailed (Figure 9c); however, marine clusters
(Clusters 1, 4, and 5) shifted towards larger aerosol sizes, suggesting aerosol-aged mixtures influencing the ability to scatter
and absorb radiation.

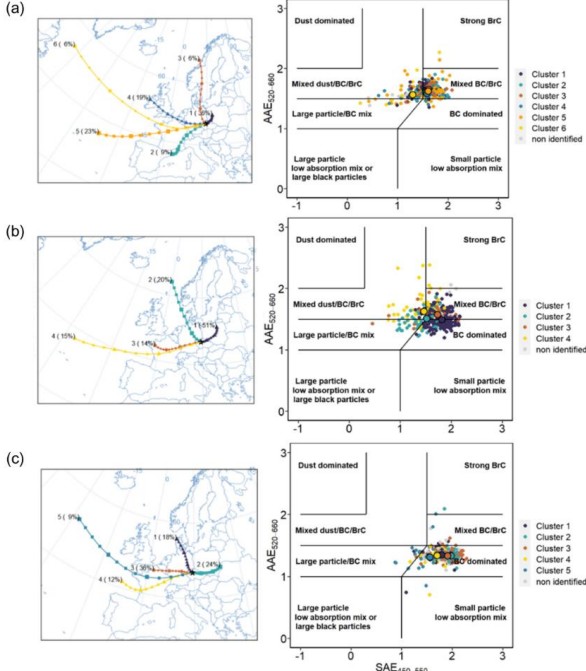

**Figure 9: The 72-h air mass back trajectories paired with $AAE_{520-660}$ vs $SAE_{450-550}$ for winter (a), spring (b), and summer (c). The plots are color-coded by trajectory cluster number, with cluster medians indicated by black circles and the percentage contribution of each cluster provided. The fall was not taken into consideration due to the limited data.**

To investigate seasonal variations in BC aerosol concentration, semi-online OC/EC measurements were used to
calculate the concentrations of the secondary (SOC) and primary (POC) organic carbon based on the method used in Mbengue
et al. (2021). The concentrations of OC, POC, SOC, and the ratio of organic and elemental carbon (OC/EC) remained stable
during winter and increased in spring, mirroring the behavior of f(RH) until May (Figure 10). However, carbonaceous
components rose from late spring into summer while mainly f(RH)$_{bsp}$ decreased, suggesting a seasonal decoupling of SOC and
aerosol hygroscopicity. This may be due to the higher fraction of POC (Liu and Wang, 2010), the formation of less hygroscopic
SOA from biogenic precursors (Huang et al., 2019), and the evaporation of semi-volatile organics at higher temperatures,
reducing their contribution to water uptake and f(RH) (Thomsen et al., 2024).





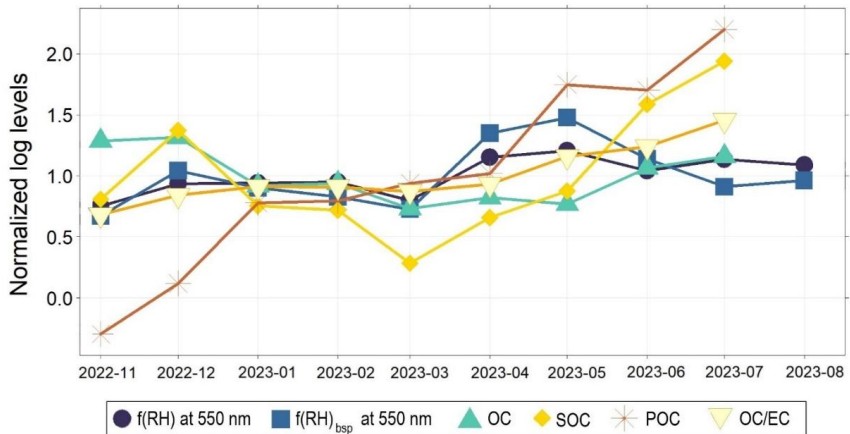

**Figure 10: Mean-normalized temporal variation of logarithm of 4-h median values of f(RH) and f(RH)$_{bsp}$, OC, SOC, POC, and OC/EC.**

Elemental carbon (EC) showed negative Spearman correlations with f(RH) and f(RH)$_{bsp}$ (R = -0.33 and -0.17), as expected. OC showed a moderate negative correlation with f(RH), yet the OC/EC ratio correlated positively with f(RH) and f(RH)$_{bsp}$, especially during spring (R = 0.31). This indicates that the relative abundance of organic carbon, rather than its absolute concentration, enhances hygroscopicity (Table S.4). The SOC/OC ratio seasonal behavior followed the OC/EC seasonality, emphasizing the role of SOA in aerosol hygroscopic growth.

The four OC subfractions distinguished by the thermo-optical method further clarified this behavior. Surprisingly, the sum of more volatile OC1 and OC2 positively correlates with f(RH). In contrast, the sum of less volatile OC3 and OC4 showed negative correlations, implying higher hygroscopicity of fresher, more volatile OC subfractions (Figure 11). This aligns with field studies of "smoke" particles by Chan et al. (2005), showing that levoglucosan from biomass burning or urban pollution can "age" into simpler dicarboxylic acids, such as succinic acid, which is less hygroscopic than the original substance, causing aged biomass burning aerosols to be less hygroscopic than the fresh ones. However, the hygroscopicity of the mentioned organic compounds highly depends on their mixing state and the additional components in the mixture (Maskey et al., 2014).

A key factor influencing this trend may be Humic-Like Substances (HULIS), typically associated with OC3 and OC4. HULIS are low in volatility and hygroscopicity (Kristensen et al., 2012) and are primarily emitted from biomass burning, residential heating, and traffic. Their presence may suppress water uptake and light scattering (Dinar et al., 2008). Urban HULIS tend to absorb light more and decrease SSA than rural sources (Tang et al., 2020). HULIS often exist in the glassy or semi-solid state, preventing diffusion of water molecules in their structure and decreasing their hygroscopicity (Koop et al., 2011), and further, aging can enhance viscosity via oligomerization, reducing hygroscopicity even more (Song et al., 2016).

The highest correlation between f(RH), f(RH)$_{bsp}$, and (OC1 + OC2)/(OC3 + OC4) ratio was found in spring (R=0.43 and 0.28) despite stronger photochemical activity during summer (Figure 11). This counterintuitive result suggests that spring conditions preserve more water-uptake-efficient OC1 and OC2 in the particulate phase, while intense summer photochemistry and higher temperatures drive oxidation and oligomerization of organic compounds, transforming initially hygroscopic semi-volatile organics (OC1+OC2) into more aged, less hygroscopic fractions (OC3+OC4), and promote volatile organics evaporation, reducing the relative contribution of fresh OC to the particulate phase.





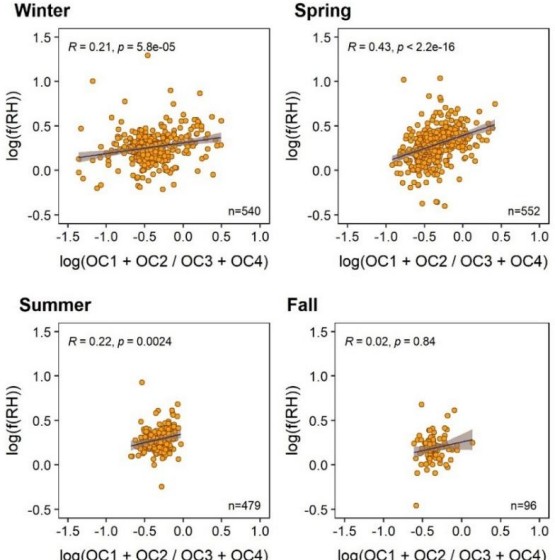

**Figure 11: Seasonal Spearman correlations between the f(RH) and the log-transformed ratio of more volatile OC fractions (OC1+OC2) and less volatile OC fractions (OC3+OC4). The values in the bottom right corner indicate the number of observations, and the values in the upper left corner indicate Spearman correlation coefficient R and the respective p values.**

### 3.6. Light scattering enhancement and NPF

Previous chapters suggested a link between light scattering enhancement and secondary particles originating from NPF, supporting the results of Liu et al. (2022) and Mitra et al. (2022) on a considerable role of SOA in the light scattering enhancement and the respective aerosol radiative impacts.

The highest number of NPF events were found in May, June, and July, correlating with the highest f(RH) in May (Figure 12). However, despite continued NPF activity, f(RH) decreased in June and July, implying a shift in aerosol chemical composition. Chemical composition estimation and absorption measurements point to traffic-related BC during summer months, which diminishes the effect of secondary particle growth on light scattering enhancement.

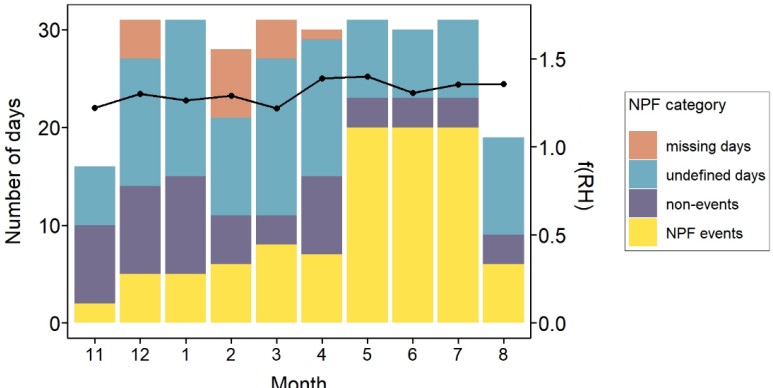

**Figure 12: Monthly NPF events classification and f(RH) median values (black line).**



The principal component analysis (PCA) was performed using meteorological variables and particle number
concentrations to investigate the role of NPF in the observed seasonal variability of light scattering enhancement. In May,
particle number concentrations in the 8–100 nm size range ($D_p\_8\_100$) strongly correlated with high global radiation (GLRD),
temperature (T), and ozone ($O_3$), conditions favorable for photochemical activity (Fig. S.10). Also PNSD analyzed during
daytime NPF events from March to August showed high 20 nm peak in May (ca. 8,000 # $cm^{-3}$, Figure 13), and the PNSD's
75th percentile shifted toward smaller diameters compared to other months, indicating more intense or more frequently initiated
NPF events, driven by favorable precursor conditions. Interestingly, August exhibited an even higher concentration of 20 nm
peak (ca. 10,000 # $cm^{-3}$) than May, but with only 10 NPF days and no clear correlation of $D_p\_8\_100$ with environmental
variables.
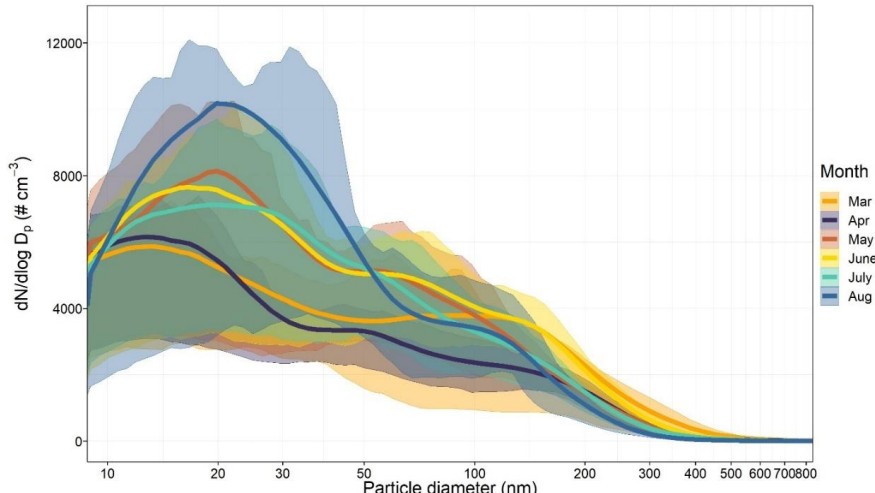
**Figure 13: The particle size number distributions during NPF days from 7:30 AM to 7:30 PM UTC.**
From diurnal analysis of PNSD in May and August, additional differences in the aerosol populations were found (Fig.
S.11). In May, morning PNSD displayed a bimodal distribution peaking at 15 nm and 60 nm, reaching ca. 5,000 and 6,000 #
$cm^{-3}$ in medians, originating from newly formed ultrafine particles and aged background particles. By midday, the distribution
was dominated by freshly nucleated particles in the 15–20 nm range, with concentrations peaking at around 12,000 # $cm^{-3}$,
shifting to ca 7,000 # $cm^{-3}$ 30 nm peak in the afternoon due to the growth and coagulation of the particles.
In August, morning PNSDs were also bimodal, with a dominating 15 nm peak (10,000 # $cm^{-3}$) and a second peak at 55
nm (6,000 # $cm^{-3}$). The peak diameter moved to 25 nm (14,000 # $cm^{-3}$) in midday and 40 nm (11,000 # $cm^{-3}$) in late afternoon.
Although August exhibited higher concentrations and stronger growth during selected NPF events, the f(RH) was lower in
August (~1.34) than in May (~1.40), indicating limited growth of ultrafine particles formed in August to optically active sizes.
On the other hand, the higher May f(RH) values reflect a combination of efficient particle growth, the higher hygroscopicity
of condensable vapors, and possibly lower background aerosol surface area, promoting nucleation and early growth stages.
These results highlight that NPF frequency, and its intensity alone do not determine optical relevance - hygroscopic growth
potential, size evolution of particles, and chemical composition of the condensing vapors are also crucial for determining
aerosol radiative impacts.



## 4. Summary and conclusions

This work presented a novel, cost-effective approach to investigate aerosol hygroscopicity using a single-nephelometer set-up with an automatically controlled switching valve alternating between humidified and dried sample branches. This design reduces uncertainties associated with dual-instrument configurations. After testing and calibrations, the system was installed at the suburban site, Suchdol in Prague, Czech Republic, from November 2022 to August 2023 to fill the knowledge gap related to the optical hygroscopic behavior of aerosol at European urban/suburban sites, particularly regarding light scattering under increased relative humidity (RH).

The light total scattering enhancement factor f(RH) and the light backscattering enhancement factor $f(RH)_{bsp}$ were derived from the humidified and dried measurement of the light total scattering ($\sigma_{sp}$) and backscattering ($\sigma_{bsp}$) coefficients at 450, 550, and 700 nm. The enhancements were analyzed in relation to climate-relevant aerosol optical properties — Scattering/Absorption Ångström Exponent SAE/AAE, hemispheric backscattering ratio $b$, asymmetry factor $g$, and Single Scattering Albedo SSA — estimated chemical composition, particle number size distributions (PNSD), meteorological parameters, back trajectory analysis, and new particle formation (NPF) events.

The measured f(RH) and $f(RH)_{bsp}$ at 550 nm were among the lowest (1.32 and 1.12 at 550 nm), together with one of the lowest $SSA_{550}$ reported in similar studies. This indicates a dominance of low-hygroscopic aerosol species originating from local combustion and traffic emissions. The seasonal and wavelength-specific differences in the relationship between $f(RH)_{bsp}$ and f(RH) underscore the importance of specific corrections in radiative models. Peaks of f(RH) and $f(RH)_{bsp}$ were detected in spring, possibly caused by the higher presence of more hygroscopic aerosol and enhanced photochemical activity. An increase of f(RH) and $f(RH)_{bsp}$ resulted in decreased $b_{550}$ and increased $g_{550}$ and $SSA_{550}$, implying enhanced total scattering and stronger forward scattering for larger particles caused by hygroscopic growth. However, $SAE_{450-700}$ increased with RH in summer, indicating shrinking of particles, possibly due to the very low-hygroscopic black carbon (BC) emissions prone to compacting upon humidification.

Chemical composition ruled the light scattering enhancement at the site more than PNSD. BC-dominated aerosol prevailed in summer, while other seasons featured aerosol mixtures with BC and BrC, occasionally dust and marine aerosol. Both f(RH) and $f(RH)_{bsp}$ correlated positively with OC/EC and SOC/OC ratios. The more volatile organic fractions exhibited a positive correlation with the light scattering enhancement compared to the less volatile ones, mainly in the spring season, highlighting the critical role of SOC in the overall aerosol hygroscopicity despite their low-hygroscopic potential, mainly at polluted sites. The principal component analysis (PCA), the PNSD, and the NPF analysis identified May as the month when NPF most strongly contributed to the overall light scattering enhancement.

In summary, the single-nephelometer system proved suitable for ambient aerosol characterization, reducing measurement uncertainty. However, the limits of this kind of measurement are the reduced time resolution, together with a lack of parallel measurement of dry and wet aerosol properties for humidogram studies. Our findings contribute to filling a key gap in the global dataset by providing insight into environments where low f(RH), low SSA, and relatively high SAE co-occur — conditions that are underrepresented in the literature.

## Authors contribution

LS, JO, PR and VZ designed the methodology of the work and created the study conceptualization. Additionally, LS was responsible for data curation, formal analysis, investigation, validation, software, visualization, and writing of the original draft. JO was responsible for software, resources, investigation, and writing — review & editing. NZ contributed to data curation, formal analysis and writing — review & editing. PR was additionally responsible for software. PV was responsible for data curation and writing — review & editing. RP was responsible for project administration, funding andministration and writing — review & editing. IH contributed to supervision. VZ was responsible for project administration, supervision, resources and writing — review & editing.



**Data availability**

Datasets including f(RH) and f(RH)$_{bsp}$, original relative humidity and temperature variation within humidified single-nephelometer set-up system, concentration of carbonaceous aerosol species, particle size number concentration, meteorological parameters, dry and humidified aerosol optical properties, NPF categorization, and air mass back-trajectory cluster identification are available at Suchánková, Lenka (2025), "Humidified single-nephelometer set-up system datasets", Mendeley Data, V1, doi: 10.17632/8ds98t2f3x.1.

**Competing interests**

The authors declare that have no competing interests.

**Acknowledgments**

This work was supported by the Ministry of Education, Youth and Sports of the CR within the Large Research Infrastructure ACTRIS Czech Republic (LM2023030) and CzeCOS (LM2023048). The authors thank the RECETOX RI (No LM2023069) for the supportive background. This work was supported by the European Union's Horizon 2020 research and innovation program under grant agreement No 857560 (CETOCOEN Excellence). This publication reflects only the author's view, and the European Commission is not responsible for any use that may be made of the information it contains. Authors acknowledge support from AdAgriF - Advanced methods of greenhouse gases emission reduction and sequestration in agriculture and forest landscape for climate change mitigation (CZ.02.01.01/00/22_008/0004635).

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
