# Peer review of "An innovative approach to measuring hygroscopic light scattering enhancement"

_EGUsphere, 2025_

## Referee Comment (RC1)

**Review of Suchánková et al.,** An innovative approach to measuring hygroscopic light scattering enhancement using a humidified single-nephelometer system

This work presented a novel, cost-effective approach to investigate aerosol hygroscopicity using a single-nephelometer set-up with an automatically controlled switching valve alternating between humidified and dried sample branches. The authors provide a succinct introduction to the single nephelometer system used to measure particle light scattering and backscattering enhancement factors f(RH), although I document some perceived issues with this design (see Suggestions 1&2 below). Results of f(RH) from Nov 2022-July 2023 were presented in the context of seasonal variability, aerosol optical properties (single scattering albedo, scattering Angstrom exponent, absorption Angstrom exponent), estimated aerosol chemistry (based on aerosol optical properties using a scheme introduced by Cappa et al., 2016), air mass source region, and OC/EC. The paper adds useful climatology of f(RH) in the Prague, Czech Republic area but I recommend some changes to improve the paper before publication, as outlined below.

**Suggestions:**

- 1. The authors claim in multiple places that the single-nephelometer aerosol scattering hygroscopicity measurement reduces uncertainties associated with dual-instrument configurations. However, no supporting evidence (error estimates) are provided to support this claim and I would argue that the single nephelometer system could potentially have larger f(RH) uncertainties than the dual nephelometer systems most commonly used. I base my statement on two potential error sources:
  - (a) Non-simultaneous sampling of dried and humidified particle light scattering: The single nephelometer system used to measure f(RH) can potentially possess smaller uncertainty relative to the traditional dual nephelometer system but switching between dried and humidified particle light scattering at 1-hour intervals can lead to potentially larger (and unknown) uncertainty in f(RH) because one is taking a ratio of measurements taken one hour apart. I should note that the authors acknowledge the limitations of non-simultaneous dry and humidified light scattering and backscattering measurements made using the single nephelometer system (lines 124-125).
  - (b) A well-calibrated dual nephelometer system includes built-in checks of nephelometer performance lacking in the single nephelometer design. This is achieved by periodically comparing light scattering and backscattering measured by both nephelometers under low RH conditions. A slope close to 1 and intercept close to zero provides high confidence in relative calibration of the nephelometers and small differences from this can be applied to correct the wet nephelometer scattering. Values of f(RH) sometimes in the 0.8-0.9 range (Fig. 8) could potentially signal small calibration drifts or other unknown uncertainties.
- 2. The calculated f(RH) should be at a stated RH and the humidified (and to less degree dried) RH values used to compute the ratio should be the same for all hours to compute meaningful statistics over the time period . For example, many studies (Burgos, et al. 2019 and some of the references there within) use  $f(RH)=\sigma_{sp}(RH=85\%)/\sigma_{sp}(RH=40\%)$ . The authors only state that humidified scattering used are when RH $\geq$ 80%. Particle scattering increases rapidly at RH near and larger than 80% so comparing multiple hours with different RH values or using the hours to calculate statistics can lead to additional, unknown uncertainties.

**Minor Suggestions**

- 1. The color-coded plots of particle type versus cluster, SSA, etc. are nicely done and convey a lot of information on single plot (Figs. 8-9).
- 2. The statement in lines 58-59 is confusing: "The comparison of the integrating nephelometer TSI 3563 with AURORA 3000 possessed an overall uncertainty of 2-5 % for σsp and 3-11 % for σbsp in laboratory conditions, respectively (Müller et al., 2011)." For which instrument is the uncertainty quoted for and (if so) is the other instrument assumed to be the "standard" reference instrument? Please clarify what you mean by these "uncertainties".
- 3. Corrections to the raw data were mentioned in lines 159-161 but no mention was made of the most important correction, namely the nephelometer angular truncation correction (Anderson and Ogren, 1998). The authors also stated that "Values below the limit of detection (LOD) were replaced by LOD/2 values" but no mention was made as to the value of the LOD.
- 4. The two wavelength pairs used to calculate  $\Delta$ SAE should be explicitly stated.
- 5. The authors state (lines 343-444) that "While we found a larger influence of the chemical composition on light scattering enhancement than the particle size, the seasonal variations in the ratio b were linked to changes in aerosol size distributions and sources." However, no data is presented to support this assertion.
- 6. Figure 6: There is a non-negligible fraction of f(RH) <1 (often for SSA<0.60) with values ~0.80-0.90 for ~5% of whole dataset. In a dual nephelometer system this could perhaps be attributed to relative differences/drifts in nephelometer calibration and this difference could be corrected by inter-comparison of the nephelometers under low RH conditions but it is not obvious to me what the reason could be for a single nephelometer system. Please elaborate on likely sources for this.

---

## Author Comment (AC1)

**Response to Reviewer #2**

**Summary:**

The manuscript presents measurements of particle hygroscopicity in the form of scattering enhancement factor derived from a single RH-switched nephelometer. A new instrument was developed and operated for approximately nine months at a suburban site outside of Prague, Czech Republic. Data are presented both for a traditional hemispheric scattering measurement and also for backscatter-specific hygroscopicity. Results suggest a dominance of low-hygroscopicity, highly-absorbing aerosol at the site, and correlations with other aerosol properties are discussed. Inclusion of backscatter f(RH) measurements is a nice unique addition to the literature and could be emphasized more throughout the manuscript. Still, major revisions are necessary before publication to address concerns regarding the validity of the approach when conditions are changing on sub-hour timescales, and the validity of highly absorbing aerosol observations.

We highly appreciate the critical insight of Reviewer #2. Changes based on the comments and suggestions of Reviewer #2 are colored blue in the manuscript and in the ´Response to the Reviewer´ document. Changes driven by Reviewer #1 are colored green in the manuscript, while other changes made are colored red.

**Major critiques:**

1. The manuscript focuses on a self-described "innovative approach" or "novel" system to measure ambient aerosol hygroscopicity (i.e., f(RH)). But in my opinion, the method that is presented is at best functionally the same as previous work, or depending on your application it is less robust. The novel aspect of the new approach is utilizing a single nephelometer and automatically switching the sample pre-treatment systematically back and forth from dry to humid relative humidity. This switching is done at 1-hour frequency, resulting in an f(RH) calculation once every 3 hours (i.e., every dry-humid-dry or humid-dry-humid cycle). For comparison, aircraft-based hygroscopicity measurements are done utilizing multiple instruments in parallel, thus producing dry and wet datasets simultaneously (i.e., Brock et al. [2015] or Ziemba et al. [2013]). Clearly, ground-based measurements do not require the same fast time-response, but the manuscript does not adequately describe the merits of using a single instrument. For example, the abstract claims (Line-17, also Line-63) that the new system "reduces measurement uncertainty" but does not justify this improvement. Can this improvement in measurement uncertainty be quantified? The authors should add text to provide more clarity. Additionally, at least one study already employs a single-nephelometer-based f(RH) measurement (Orozco et al., 2016) and should be cited.

The presented single-nephelometer system serves as an alternative to the two-nephelometer system for studies of aerosol hygroscopicity. It was designed to be cost-effective, suitable for long-term observations, and to reduce instrumental noise in the derived backscattering enhancement. By relying on a single instrument, uncertainties associated with two independent nephelometers do not propagate into the ratio, resulting in an improved signal-to-noise ratio compared to a two-nephelometer configuration. When it comes to the question of uncertainty, we primarily referred to the reduction of uncertainty in the f(RH) estimation that arises from comparing two instruments, each with its own set of uncertainties, as the presented approach utilizes only one instrument. As a result, this type of uncertainty practically diminishes. However, authors agree that the presented approach is less robust than some previously reported

studies due to hourly alternation between dry and humidified mode. We were not aware of the paper of Orozco et al. (2016), thank you for bringing this to our attention. As a result, we added the citation of Orozco et al. (2016) and reconsidered the description of the presented system:

Lines 17-19: "This study presents a single-nephelometer system that reduces measurement uncertainty associated with inter-instrument comparison and enables the study of aerosol hygroscopic behavior in the inadequately represented European urban environment."

Lines 64-70: "Thus, this study presents a single-nephelometer system to partially reduce uncertainties in the f(RH) estimation arising from the comparison of two instruments and to investigate ambient aerosol particles' light scattering hygroscopic behavior at the suburban site. Orozco et al. (2016) examined aerosol hygroscopicity using a dryer-humidifier system coupled to a TSI 3563 nephelometer in urban/suburban environments in North America. However, to the best of our knowledge, only one study has specifically investigated aerosol light-scattering enhancement in a European urban/suburban environment (Titos et al., 2014). Therefore, this study provides a unique insight into f(RH) and light-scattering enhancement of aerosols in a European suburban context using a single-nephelometer approach."

Lines 133-136: "While this approach reduces uncertainties in the f(RH) estimation arising from the comparison of two instruments, it should be noted that such a measurement approach relies on the reduced time resolution compared to the dual nephelometer setups (1-hour intervals), which may introduce additional uncertainty on short timescales, e.g., the influence of episodic extreme pollution events."

Lines 521-524: "This work presented a cost-effective approach to investigate aerosol hygroscopicity using a single-nephelometer set-up with an automatically controlled switching valve alternating between humidified and dried sample branches. This design reduced uncertainties associated with dual-instrument configurations while allowing for investigation of light backscattering changes with elevated RH."

While the single-nephelometer system has been already successfully applied in Orozco et al. (2016), such an approach generally relies on the assumption that ambient aerosol properties do not vary substantially over the dry–humid switching period. At the present suburban site, rapid temporal variability in aerosol loading and meteorological conditions was frequently observed. To mitigate potential biases associated with this variability, an overlapping-ratio approach was applied in the data analysis.

2. Another result of this work is the dominance of low-hygroscopicity and highly-absorbing aerosol in the region. Table 1 reports seasonal SSA values of 0.65-0.76. These are extremely low for ambient aerosols and require more explanation. For example, SSA values globally are typically greater than 0.9 (Devi and Satheesh, 2022), even in regions dominated by biomass burning. Part of this concern is the lack of discussion regarding aethalometer-based absorption corrections (of which SSA values are based). Aethalometer observations are filter-based, and typically require correction for scattering from the filter media directly or from the aerosols collected on the filter (e.g., Virkkula et al. [2010]). Backman et al. [2017] report a correction factor of 3.45 for Arctic sampling. Coen et al. [2010] also describe numerous correction schemes developed for aethalometers. Given such low SSA values, a more in-depth discussion and scrutiny of the reported absorption data is critically important.

Based on this comment, a mistake was found in the calculation of aerosol absorption properties, leading to a significant error in SSA estimation – thank you for noticing this! Now, the SSA was recalculated, and the revised results were implemented throughout the manuscript and Supplementary Materials, including Table 1, Figures 6, 7, 8, S.5, S.7, S.8, and S.9. All modifications are highlighted in red.

The details regarding absorption corrections used in data analysis were added to the Chapter Materials and Methods. More details are provided in the section "Specific Comments" below.

Although the revised SSA values (median of 0.81) are still lower than the global mean values reported by Devi and Satheesh (2022), it should be noted that their estimates are based on satellite-derived, column-integrated SSA, whereas the present study relies on in situ, near-surface measurements. Such in situ observations are more sensitive to locally emitted, freshly produced absorbing aerosols, particularly in urban and suburban environments. Titos et al. (2021) reported SSA values below 0.9 at several continental stations, especially those influenced by traffic and fossil-fuel combustion. The SSA values observed in this study are consistent with the range reported for such environments, indicating that they are physically plausible and representative of the local aerosol source mixture.

**Specific Comments:**

TITLE     I'm not sure that "innovative approach" should be highlighted in the title, since the I don't think the method is really the focus of the paper. I suggest revising to something emphasizing seasonal variability for hemispheric and backscatter hygroscopicity.

We reconsidered the manuscript title as suggested by Reviewer #2 and changed it to:
"Hygroscopic enhancement of suburban aerosol light scattering measured using a single-nephelometer system in Central Europe"

74        How is instrument exhaust treated and are you able to verify that exhaust never contaminates the sample line?

Due to the insufficient efficiency of the internal pump of the TSI 3563, experienced during the permanent installation as the ACTRIS instrument at the National Atmospheric Observatory Košetice, it was replaced by an external pump with an efficient HEPA filter to capture the exhaust particles and water molecules. The filter's efficiency was regularly checked, and it was replaced when needed.
Moreover, the four-way switching valve alternating between dried and humidified air (Fig. 2, No.5) ensured that always one type of the sample was directed to the nephelometer while the second type of sample was carried by the additional tubing out of the sampling container, away from the sampling head.

85        Please add additional instrumental details for the instrument, including:

  - Nafion dryer model number for the common dried line
  - Nafion dryer model for the humidifier stage,

The model numbers for dried line and humidifying line (MD-700-24), plus the bundle Nafion membrane responsible for the particle-free air humidification (FC100) were added to the MS (Lines 91-97 and 101-102, respectively).

- theoretical (or preferably experimental) size-dependent particle losses in each of the Nafion dryers,
- theoretical (or preferably experimental) size-dependent particle losses in the 4-way valve, and

Considering the vertical orientation of the Nafion membranes, particle losses are expected to be dominated by Brownian diffusion. Such diffusion losses primarily affect ultrafine particles (< 100 nm) and are therefore considered negligible for aerosol light-scattering measurements. The TSI 3563 nephelometer operates down to a wavelength of 450 nm, for which optically relevant scattering is dominated by particles larger than approximately 100–200 nm, whereas particles below ~100 nm contribute negligibly to the measured scattering signal.

In the case of the 4-way valve, particle losses may occur due to inertial impaction in the bends of the valve, particularly for humidified particles that grow in size compared to their dried counterparts. We calculated theoretical losses for $PM_{10}$ in the 4-way valve as an upper bound (~ 5.2 %). While urban air rarely contains significant numbers of particles larger than 2–3 µm away from direct dust or road sources, these particles contribute disproportionately to the scattering signal due to their volume (Held et al., 2008; Klejnowski et al., 2013; Wu and Boor, 2021). According to the classical aerosol size distribution model of Junge (1995), particle number decreases roughly as a power law with increasing diameter, providing a theoretical explanation for the scarcity of coarse particles. For particles < 2.5 µm and even < 1 µm , which dominate urban aerosol volume and scattering (Salam et al., 2012; Wu and Boor, 2021), losses in the valve are minimal (~ 0.3 % and ~ 0.03 %, respectively), indicating that the instrument effectively captures the relevant fraction of the aerosol.
Consequently, neither Nafion- nor valve-related particle losses are expected to significantly influence the reported scattering results.

The information regarding expected losses was added into the manuscript (lines 129-132): "Particle losses in the Nafion membranes and the 4-way valve are not expected to significantly influence the reported scattering measurements, as diffusional losses affect optically negligible ultrafine particles and inertial losses primarily affect coarse particles that are scarce in urban air. Urban aerosol scattering is dominated by accumulation-mode particles (<1 µm), for which calculated valve losses are minimal."

- typical water temperature used for humidification, and whether there is any heating of sample air in the system

The water in the thermostat was heated up between 31 and 33 °C. No additional heating was applied in the experimental design. Additionally, the sampling lines carrying the humidified aerosol sample were insulated with insulation foam. The protective shield from the TSI 3563 was removed during the campaign, and the halogen lamp illuminating the measurement cell was constantly cooled by the cooling fan to ensure minimal temperature increase in the measuring cell. The information was added to lines 100-103: "Demineralized water, heated in a controlled manner by the thermostat (up to 33 °C), was directed to the bundle Nafion membrane (FC100, Permapure) (Fig. 2, No. 7), where mass transfer between water (in channels) and the dry particle-free air (outside the channels) occurred."

145    Please provide a description and references for any corrections applied to the aethalometer data.

The information was added to the manuscript (lines 160-173):
"A dual-spot multiwavelength aethalometer (Model AE33, Magee Scientific, USA, 2018) continuously measured light attenuation by particles at seven wavelengths (370, 470, 520, 590, 660, 880, and 950 nm). The dual-spot technology enables real-time compensation for filter loading. Particles were sampled through the $PM_{10}$ sampling head (Leckel GmbH) at a flow rate of 5 lpm, dried in a custom-made Nafion dryer (TROPOS, Leipzig, Germany), and deposited onto tetrafluoroethylene (TFE) coated glass filter tape. Light transmission through the deposited sample is measured and compared to the blank filter tape spot as a reference, converting the optical absorbance into an equivalent black carbon concentration (eBC, $\mu g\ m^{-3}$) data. The data was automatically corrected by the multi-scattering correction factor C (1.39 for the recommended filter tape M8060). Furthermore, the wavelength-dependent mass absorption cross-section (MAC) values were used for the eBC conversion to the absorption coefficients $\sigma_{ap}$ (Drinovec et al., 2015; Müller and Fiebig, 2021; Savadkoohi et al., 2025). The wavelength-dependent MAC factors were adopted from the AE33 manual (e.g., MAC = 7.77 $m^2$/g for 880 nm) (Magee Scientific, 2018). The $\sigma_{ap}$ values were additionally standardized to STP conditions (273.15 K, 1013.25 hPa) and divided by the harmonization factor H* (1.76 for the recommended filter tape M8060), which compensates for the differences between the predefined multi-scattering correction factor C and corrections in the Aethalometer firmware set by the manufacturer (Müller and Fiebig, 2021; Savadkoohi et al., 2024, 2025)."

150    Please provide the mass absorption efficiency used to convert absorption to BC mass. Was this conversion factor wavelength dependent?

Yes, the mass-absorption cross-section (MAC) factor used for the eBC-to-$\sigma_{ap}$ recalculation was wavelength-dependent, adopted from the AE33 manual. Specifically, the following MAC values were used (Magee Scientific, 2018):

| Wavelength (nm) | MAC ($m^2\ g^{-1}$) |
| --- | --- |
| 370 | 18.47 |
| 470 | 14.54 |
| 520 | 13.14 |
| 590 | 11.58 |
| 660 | 10.35 |
| 880 | 7.77 |
| 950 | 7.19 |

The information regarding MAC was added to the manuscript (lines 167-169):
"Furthermore, the wavelength-dependent mass absorption cross-section (MAC) values were used for the eBC conversion to the absorption coefficients $\sigma_{ap}$ (Drinovec et al., 2015; Müller and Fiebig, 2021; Savadkoohi et al., 2025). The wavelength-dependent MAC factors were adopted from the AE33 manual (e.g., MAC = 7.77 $m^2$/g for 880 nm) (Magee Scientific, 2018)."

167    Why do you have to assume the dew-point is preserved when you are directly measuring both the inlet and outlet temperature and RH? From that data, did you have to filter any of the dataset for instances when the RH dramatically changed inside the nephelometer?

We appreciate this on-point question. Temperature and RH data from all sensors were corrected using calibration curves derived from comparisons with a reference thermometer and a dewpoint mirror hygrometer. Although RH and temperature varied along the nephelometer flow path, the water vapor mixing ratio is conserved under steady-state conditions in the absence of the water phase changes (condensation or evaporation). Therefore, the real RH of the sample was derived by assuming identical dew point temperatures in front of and behind the measurement cell, following the methodology of Ren et al. (2021). Dew point temperatures were independently calculated from the inlet and outlet RH/T measurements using the Magnus–Tetens approximation (Alduchov and Eskridge, 1997) (See Chapter 2.4.1. Humidified nephelometer system in the manuscript).

Regarding data filtering, switching between humidified and dried modes leads to transient non-equilibrium conditions.

Comparing inlet- and outlet-derived dew point temperatures enabled us to identify periods when RH equilibrium was not achieved inside the nephelometer. Based on this analysis, approximately 15 % of the data were excluded due to significant RH differences before and after the measurement cell. The lines 203-204 were slightly updated: "Data corresponding to the range of 40 % < RH < 80 %, including the conditioning periods, was discarded (approx. 15 % of the raw data)."

180      When enhancement factors are calculated, are they referenced specifically to RH = 40% and RH = 80%? If the RH control varied or drifted, was the data corrected back to 40 and 80% for calculation of the enhancement factor? For example, if the dry RH was actually controlled at 32%, was this scattering data corrected to 40% or simply assumed to be "dry"? Similarly for the humidified sample, how did you treat data when the control RH deviated from 80%?

This comment matches the concern of Reviewer #1. The enhancement factors were calculated at $RH \geq 80$ % vs $RH \leq 40$ %. Despite the efforts, the RH of the sample in both modes fluctuated throughout the measurement campaign, as shown in Table S.2 of the Supplementary Materials.

Table S.2: The RH (%) and T (K) statistics in the cell during humidified and dry modes. P10, P25, P50, P75, and P90 denote respective percentiles.

| | Humidified mode | | Dry mode | |
|---|---|---|---|---|
| | $RH_{cell}$ (%) | $T_{cell}$ (K) | $RH_{cell}$ (%) | $T_{cell}$ (K) |
| **Mean ± sd** | 87.34 ± 2.67 | 297.91 ± 4.15 | 28.36 ± 5.85 | 297.94 ± 4.16 |
| **P10** | 83.62 | 293.57 | 20.30 | 293.46 |
| **P25** | 85.81 | 295.01 | 25.05 | 294.92 |
| **P50** | 87.64 | 296.83 | 28.84 | 296.77 |
| **P75** | 89.31 | 299.13 | 32.43 | 299.29 |
| **P90** | 90.50 | 304.94 | 35.34 | 304.93 |

We are aware of this limitation; unfortunately, the single-nephelometer setup is unable to perform humidogram studies, which are essential for possible recalculation of f(RH) to a precise RH.

To improve the clarity of the text, the description of this limitation was subsequently incorporated into the uncertainty description in lines 136-140: "Moreover, another limitation of this setup originates from a lack of parallel measurement of dry and wet aerosol properties, which rules out the hygroscopic scanning, humidogram analyses, and the precise recalculation of f(RH) at the given RH. The statistical overview of RH and temperature during the measurement campaign for both humidified and dry modes is shown in Table S.2 in Supplementary Materials."

188 Are the conditioning periods shown in Figure 3? Can those periods be marked in the figure to assess stability in the system?

An example of the conditioning and measurement periods has been integrated into Figure 3 in the MS (below). Lines 200-205 were updated accordingly: "The $RH_{cell}$ dataset was later combined with the nephelometer and switching valve data. The $RH_{cell}$ data, along with the valve position dataset, were used to separate the data into dry (RH ≤ 40 %) and humidified (RH ≥ 80 %) datasets. They were also used to separate the conditioning periods, which occurred when switching between dry and humidified modes to reach RH equilibrium in the measurement cell from the actual measurement periods (Fig. 3). Data corresponding to the range of 40 % < RH < 80 %, including the conditioning periods, was discarded (approx. 15 % of the raw data). The dry and humidified datasets of aerosol light scattering properties were averaged hourly."

[Figure]

Figure 3: The example of f(RH) calculation from a single-nephelometer measurement on December 12, 2022. The D and H symbols indicate "dry" and "humid" measuring intervals of $\sigma_{sp}$. The orange intervals on the left represent the conditioning period, while the purple intervals represent the actual measurements.

188 This looks like a rather ideal period for calculating the 50-minute averages, but how does your method handle periods when dry scattering changes significantly during the 3-hour period? Gradual changes, but more likely fast changes associated with frontal passages or airmass changes, will could result in incorrect f(RH) calculations. How is this flagged or filtered in your method?

We acknowledge the critical insight of Reviewer #2 into the method and are aware of the limitation regarding the hourly averaging of measurement intervals. The most suitable solution to this issue was the use of the overlapping dry-centered and humid-centered interval calculation to capture potential fast changes in the aerosol origin or meteorological conditions. Moreover, the dry and humid light scattering properties were inspected manually alongside the particle number concentration beforehand the actual f(RH) and f(RH)$_{bsp}$ calculation. In case of a sudden significant change in the aerosol dry scattering, such interval was inspected and discarded if needed.

235 How are new particle formation events relevant to this work? This section seems outside the scope of the paper and should be removed.

After the consideration, we decided to follow Reviewer #2´s advice and removed Chapter 3.6 Light scattering enhancement and NPF from the manuscript. The Abstract, together with the Summary and conclusions, was edited accordingly.

288      It's not clear that f(RH) truly "varies with seasons", since only Fall is inconsistent. Likewise, it is very difficult to assess whether the different Fall slope is real or just a statistical anomaly (the yellow points are very hard to see). You may want to consider 4 separate panels, one for each season.

Based on this recommendation, new Figure 4 was prepared, and lines 304-305 were added to soften the statement about seasonal variations of bivariate f(RH) vs f(RH)$_{bsp}$ fit:
"Despite the strong linear relationship, f(RH)$_{bsp}$ does not precisely mirror f(RH) and slightly varies mainly between colder (fall and winter) and warmer (spring and summer) seasons."

[Figure]

Figure 4: The weighted bivariate fit of f(RH)$_{bsp}$ vs the f(RH) at $\lambda$ = 550 nm for individual seasons (left). The right plot demonstrates the weighted bivariate fit for the whole dataset. The grey points represent the overall f(RH) and f(RH)$_{bsp}$ mean value with error bars.

284      It appears that some fraction of datapoints show f(RH) and f(RH)bsp values below 1. Interestingly, the data do not converge at f(RH) AND f(RH)$_{bsp}$ = 1, with potentially more backscatter data below 1. Can you comment on the interpretation of these sub-1 data, and why backscatter might behave differently? Do you think a soot restructuring process is occurring similar to Shingler et al. [2016]?

We interpret values of f(RH) and f(RH)$_{bsp}$ < 1 mainly as measurement artefacts inherent to humidified nephelometer systems rather than a physical aerosol property. Even at sufficiently high RH, aerosol particles may not instantaneously reach equilibrium with water vapor, particularly during transitions between dry and wet sampling and under rapidly changing ambient conditions. In addition, hygroscopic growth shifts particles toward larger sizes, enhancing forward scattering and thereby increasing angular truncation effects in the nephelometer, which can partially offset the true scattering enhancement. Given that the overall f(RH) and f(RH)$_{bsp}$ at the Suchdol site were relatively low compared to other locations (e.g., Titos et al., 2021), the RH-induced change in scattering may be small relative to instrumental uncertainty. Under these conditions, exact convergence of f(RH) and f(RH)$_{bsp}$ to unity is not expected.

The authors assume that a higher occurrence of f(RH)$_{bsp}$ < 1 compared to f(RH) is consistent with the overall larger uncertainty of backscattering coefficient measurement by integrating nephelometers, for which the signal-to-noise ratio is inherently lower. Müller et al. (2011) showed that, after correction for angular truncation errors, total scattering coefficients ($\sigma_{sp}$)

measured by the Ecotech Aurora 3000 differ from those measured by a reference TSI 3563 by 2–5%, whereas backscattering coefficients ($\sigma_{bsp}$) differ by 1–11% under laboratory conditions. Moreover, because hygroscopic growth preferentially enhances forward scattering, small RH-induced changes may increase $\sigma_{sp}$ while leaving $\sigma_{bsp}$ unchanged or even reduced within uncertainty, leading to a higher frequency of $f(RH)_{bsp} < 1$.

Considering findings in Shingler et al. (2016), we performed an analysis analogous to the one in Figure 8, for data corresponding to $f(RH) < 1$ and $f(RH)_{bsp} < 1$ (Figure R.1 and R.2, see below). Sub-1 enhancement factors were observed throughout the year. In winter, the highest frequency of $f(RH) < 1$ coincided with aerosol optical properties indicative of mixed BC/BrC influence, suggesting a non-negligible contribution from biomass burning sources. We hypothesized that soot restructuring occurred in summer due to the dry and humidified $SAE_{450-700}$ behavior and the overall BC-dominated aerosol mixture (Figure 7 and Figure R.1). Shingler et al. (2016) reported the observations of $f(RH)$ and $g(RH) < 1$ in association with particles enriched in carbonaceous material, particularly during biomass burning or smoke-influenced periods. They also noted that soot restructuring may occur even when $f(RH)$ and $g(RH)$ are both > 1, highlighting that the underlying mechanisms remain poorly understood. In this context, while measurement uncertainty likely explains most occurrences of $f(RH)$ and $f(RH)_{bsp} < 1$ at the Suchdol site, the seasonal dependence and association with BC-dominated aerosol in summer suggest that soot restructuring cannot be ruled out under specific conditions.

[Figure]

Figure R.1 $AAE_{520-660}$ vs. $SAE_{450-550}$ hourly means color-coded with $SSA_{550}$ with the aerosol characterization matrix for fall, winter, spring, and summer, adopted from Cappa et al. (2016) and Cazorla et al. (2013). The red circle with red error bars estimated the median $AAE_{520-660}$-$SAE_{450-550}$ point with interquartile ranges. The circle color is coded by the median of $SSA_{550}$. The data corresponds to a timeline where $f(RH) < 1$. n identifies the number of data points in the respective season.

[Figure]

Figure R.2: $AAE_{520-660}$ vs. $SAE_{450-550}$ hourly means color-coded with $SSA_{550}$ with the aerosol characterization matrix for fall, winter, spring, and summer, adopted from Cappa et al. (2016) and Cazorla et al. (2013). The red circle with red error bars estimated the median $AAE_{520-660}$-$SAE_{450-550}$ point with interquartile ranges. The circle color is coded by the median of $SSA_{550}$. The data corresponds to a timeline where $f(RH)_{bsp} < 1$. A symbol "n" identifies the number of data points in the respective season.

303    The slopes from Figure 4 seem very different from the values stated here. Can you comment on the cause of the difference?

The discrepancy between reported values arises because the regression slopes shown in Figure 4 and the percentage differences reported in the text quantify different aspects of the relationship between $f(RH)_{bsp}$ and $f(RH)$.
The slopes represent the covariation between $f(RH)_{bsp}$ and $f(RH)$ and are influenced by the non-zero intercept and the full distribution of paired values. In contrast, the percentage differences were derived from ratios of median values and therefore reflect typical magnitude differences under representative high-RH conditions. As a result, the regression slopes should not be interpreted as direct percentage differences between $f(RH)_{bsp}$ and $f(RH)$. We clarified this point in the revised manuscript (lines 310-312): "These regression parameters describe the co-variation between $f(RH)_{bsp}$ and $f(RH)$, whereas magnitude differences between the two enhancement factors are quantified below using median values."

411    This discussion of hygroscopicity variability as a function of OC/POC/SOC is challenging without knowledge of the aerosol sulfate content. Could most of the f(RH) variability be driven by the organic mass fraction, and be less sensitive to relative contributions

of different organic species? Please comment on the importance of sulfate (or nitrate) for interpreting hygroscopicity.

We fully acknowledge the limitations of this study, in particular the absence of direct measurements of inorganic aerosol components such as sulfate and nitrate at the Suchdol site. The site is not equipped with online chemical composition instrumentation (e.g., ACSM or AMS), which prevents a full assessment of local aerosol hygroscopicity. We recognize the dominant role of inorganic salts, especially sulfate and nitrate, in driving aerosol hygroscopic growth and optical enhancement, as widely documented in previous studies (Andrews et al., 2021; Kang et al., 2025; Pöhlker et al., 2023; Titos et al., 2014).

Consistent with previous humidified nephelometer studies, f(RH) is often negatively correlated with the total organic mass fraction, reflecting low hygroscopicity of ambient organic aerosol compared to inorganic salts (Li et al., 2025; Massoli et al., 2009; Ren et al., 2021). However, this relationship is not universal and can depend on aerosol mixing state and organic chemical characteristics. While the influence of individual organic species on particle hygroscopicity has been more extensively investigated using HTDMA and CCN techniques (Han et al., 2022; Rickards et al., 2013; Suda et al., 2014), humidified nephelometer studies typically parameterize organic aerosol as a bulk mass component, with limited consideration of organic fractionation (Ren et al., 2021; Wu et al., 2017; Zieger et al., 2014).

In this context, our analysis does not aim to isolate the absolute contribution of organic fractions to f(RH), but rather to examine whether separating OC into POC and SOC provides additional explanatory power for variability in optical hygroscopicity beyond total organic mass, mainly in carbon-rich environment. The results suggest that different organic fractions are associated with systematically different f(RH) responses, consistent with previous findings indicating a stronger hygroscopic influence of secondary organic aerosol compared to primary organic material (Kuang et al., 2021). This supports the interpretation that organic aerosol cannot always be treated as a chemically uniform component when interpreting optical hygroscopic growth.

To fully acknowledge the significant role of inorganic salts in the aerosol hygroscopicity, further explanation was added to lines 463-472: "Consistent with previous hygroscopic studies, aerosol hygroscopic changes are often negatively correlated with the organic mass fraction, reflecting the low hygroscopicity of ambient organic aerosol compared to strongly hygroscopic inorganic salts (mainly sulphates and nitrates) (Andrews et al., 2021; Kang et al., 2025; Massoli et al., 2009; Pöhlker et al., 2023). We note that direct measurements of inorganic aerosol components were not available at the Suchdol site, and therefore their contributions to f(RH) variability cannot be quantified in this study. Nevertheless, a substantial fraction of the variability in f(RH) is likely controlled by variations in the inorganic fraction, and correlations between f(RH) and organic metrics can partly reflect this co-variability. Separating OC into primary (POC) and secondary (SOC) fractions allows investigation of differences in effective hygroscopicity within the organic aerosol itself. To further explore variability in optical hygroscopicity beyond total organic mass, semi-online OC/EC measurements were used to calculate the concentrations of secondary (SOC) and primary (POC) organic carbon, following the method of Mbengue et al. (2021). "

424    Similar comment.  Most literature shows that hygroscopicity increases as organic mass fraction decreases (contrary to your statement), e.g., Massoli et al. [2009].  Please comment.

The authors are aware that most studies report an increase in aerosol hygroscopicity as the inorganic fraction increases and the organic fraction decreases, consistent with the generally lower hygroscopicity of organic aerosol compared to inorganic salts, as previously mentioned (Li et al., 2025; Massoli et al., 2009; Ren et al., 2021; Zieger et al., 2014). Direct measurements of inorganic aerosol components were not available at the Suchdol site, preventing a quantitative assessment of their influence on f(RH). In carbon-rich environments like Suchdol, differences in organic chemical composition, particularly between primary (POC) and secondary (SOC) organic carbon, can significantly influence hygroscopicity. Aged, oxidized SOC is more water-soluble and typically more hygroscopic than POC (Kuang et al., 2021; Wei et al., 2024), which may contribute to observed variations in f(RH) even in the absence of inorganic composition data. Moreover, the results of this study still showed a negative correlation of OC with f(RH), which is consistent with the previous studies, and referred to the ratios of OC/EC or SOC/OC as potential factors influencing hygroscopicity at the site.
Lines 458-461 were updated in the manuscript to decrease potential confusion of the reader: "Consistent with previous hygroscopic studies, aerosol hygroscopic changes are often negatively correlated with the organic mass fraction, reflecting the low hygroscopicity of ambient organic aerosol compared to strongly hygroscopic inorganic salts (mainly sulphates and nitrates) (Andrews et al., 2021; Kang et al., 2025; Massoli et al., 2009; Pöhlker et al., 2023).

420    The y-axis of this plot is difficult to understand.  Can you update the axis label to be more explicit with how you calculated "normalized log levels"?

Figure 10 and its caption were updated:

[Figure]

Figure 10: Temporal variation of logarithmic median values of f(RH) and f(RH)$_{bsp}$, OC, SOC, POC, and OC/EC. Each series was divided by its mean value to compensate for different numeric scales.

451     Again, NPF correlation does not seem causal or robust, and should be removed, including Figures 12 and 13.

After the consideration, the authors decided to follow Reviewer #2´s advice and removed Chapter 3.6 Light scattering enhancement and NPF from the manuscript as in the case of the previous comment. The Abstract, together with the Summary and conclusions were edited accordingly.

**References used in the Response to Reviewer #2:**

Alduchov, O. A. and Eskridge, R. E.: Magnus-Tetens formula, Asheville, 21 pp., https://doi.org/https://doi.org/10.2172/548871, 1997.

Andrews, E., Zieger, P., Titos, G., and Burgos, M.: Evaluation and improvement of the parameterization of aerosol hygroscopicity in global climate models using in-situ surface measurements (Final Report), https://doi.org/10.2172/1706478, 2021.

Cappa, C. D., Kolesar, K. R., Zhang, X., Atkinson, D. B., Pekour, M. S., Zaveri, R. A., Zelenyuk, A., and Zhang, Q.: Understanding the optical properties of ambient sub-and supermicron particulate matter: Results from the CARES 2010 field study in northern California, Atmos. Chem. Phys., 16, 6511–6535, https://doi.org/10.5194/acp-16-6511-2016, 2016.

Cazorla, A., Bahadur, R., Suski, K. J., Cahill, J. F., Chand, D., Schmid, B., Ramanathan, V., and Prather, K. A.: Relating aerosol absorption due to soot, organic carbon, and dust to emission sources determined from in-situ chemical measurements, Atmos. Chem. Phys., 13, 9337–9350, https://doi.org/10.5194/acp-13-9337-2013, 2013.

Devi, A. and Satheesh, S. K.: Global maps of aerosol single scattering albedo using combined CERES-MODIS retrieval, Atmos. Chem. Phys., 22, 5365–5376, https://doi.org/10.5194/acp-22-5365-2022, 2022.

Drinovec, L., Močnik, G., Zotter, P., Prévôt, A. S. H., Ruckstuhl, C., Coz, E., Rupakheti, M., Sciare, J., Müller, T., Wiedensohler, A., and Hansen, A. D. A.: The "dual-spot" Aethalometer: An improved measurement of aerosol black carbon with real-time loading compensation, Atmos. Meas. Tech., 8, 1965–1979, https://doi.org/10.5194/AMT-8-1965-2015, 2015.

Han, S., Hong, J., Luo, Q., Xu, H., Tan, H., Wang, Q., Tao, J., Zhou, Y., Peng, L., He, Y., Shi, J., Ma, N., Cheng, Y., and Su, H.: Hygroscopicity of organic compounds as a function of organic functionality, water solubility, molecular weight, and oxidation level, Atmos. Chem. Phys., 22, 3985–4004, https://doi.org/10.5194/ACP-22-3985-2022, 2022.

Held, A., Zerrath, A., McKeon, U., Fehrenbach, T., Niessner, R., Plass-Dülmer, C., Kaminski, U., Berresheim, H., and Pöschl, U.: Aerosol size distributions measured in urban, rural and high-alpine air with an electrical low pressure impactor (ELPI), Atmos. Environ., 42, 8502–8512, https://doi.org/10.1016/j.atmosenv.2008.06.015, 2008.

Junge, C.: THE SIZE DISTRIBUTION AND AGING OF NATURAL AEROSOLS AS DETERMINED FROM ELECTRICAL AND OPTICAL DATA ON THE ATMOSPHERE, J. Meteorol., 12, 13–25, https://doi.org/https://doi.org/10.1175/1520-0469(1955)012<0013:TSDAAO>2.0.CO;2, 1995.

Kang, H., Jung, C. H., Lee, B. Y., Krejci, R., Heslin-Rees, D., Aas, W., and Yoon, Y. J.: Aerosol hygroscopicity influenced by seasonal chemical composition variations in the Arctic region, J. Aerosol Sci., 186, 106551, https://doi.org/10.1016/j.jaerosci.2025.106551, 2025.

Klejnowski, K., Krasa, A., Rogula-Kozłowska, W., and Błaszczak, B.: Number size distribution of ambient particles in a typical urban site: The first polish assessment based on long-term (9 months) measurements, Sci. World J., 2013, https://doi.org/10.1155/2013/539568, 2013.

Kuang, Y., Huang, S., Xue, B., Luo, B., Song, Q., Chen, W., Hu, W., Li, W., Zhao, P., Cai, M., Peng, Y., Qi, J., Li, T., Wang, S., Chen, D., Yue, D., Yuan, B., and Shao, M.: Contrasting effects of secondary organic aerosol formations on organic aerosol hygroscopicity, Atmos. Chem. Phys., 21, 10375–10391, https://doi.org/10.5194/ACP-21-10375-2021, 2021.

Li, L., Li, M., Fan, X., Chen, Y., Lin, Z., Hou, A., Zhang, S., Zheng, R., and Chen, J.:

Measurement report: The variation properties of aerosol hygroscopic growth related to chemical composition during new particle formation days in a coastal city of Southeast China, Atmos. Chem. Phys., 25, 3669–3685, https://doi.org/10.5194/ACP-25-3669-2025, 2025.

Magee Scientific: User's manual for Aethalometer® Model AE33, 149, 2018.

Massoli, P., Bates, T. S., Quinn, P. K., Lack, D. A., Baynard, T., Lerner, B. M., Tucker, S. C., Brioude, J., Stohl, A., and Williams, E. J.: Aerosol optical and hygroscopic properties during TexAQS-GoMACCS 2006 and their impact on aerosol direct radiative forcing, J. Geophys. Res. Atmos., 114, 0–07, https://doi.org/10.1029/2008JD011604, 2009.

Mbengue, S., Zikova, N., Schwarz, J., Vodička, P., Šmejkalová, A. H., and Holoubek, I.: Mass absorption cross-section and absorption enhancement from long term black and elemental carbon measurements: A rural background station in Central Europe, Sci. Total Environ., 794, 1–14, https://doi.org/10.1016/j.scitotenv.2021.148365, 2021.

Müller, T. and Fiebig, M.: ACTRIS In Situ Aerosol: Guidelines for Manual QC of AE33 absorption photometer data, 9 pp., 2021.

Müller, T., Laborde, M., Kassell, G., and Wiedensohler, A.: Design and performance of a three-wavelength LED-based total scatter and backscatter integrating nephelometer, Atmos. Meas. Tech., 4, 1291–1303, https://doi.org/10.5194/amt-4-1291-2011, 2011.

Orozco, D., Beyersdorf, A. J., Ziemba, L. D., Berkoff, T., Zhang, Q., Delgado, R., Hennigan, C. J., Thornhill, K. L., Young, D. E., Parworth, C., Kim, H., and Hoff, R. M.: Hygrosopicity measurements of aerosol particles in the San Joaquin Valley, CA, Baltimore, MD, and Golden, CO, J. Geophys. Res. Atmos., 121, 7344–7359, https://doi.org/10.1002/2015JD023971, 2016.

Pöhlker, M. L., Pöhlker, C., Quaas, J., Mülmenstädt, J., Pozzer, A., Andreae, M. O., Artaxo, P., Block, K., Coe, H., Ervens, B., Gallimore, P., Gaston, C. J., Gunthe, S. S., Henning, S., Herrmann, H., Krüger, O. O., McFiggans, G., Poulain, L., Raj, S. S., Reyes-Villegas, E., Royer, H. M., Walter, D., Wang, Y., and Pöschl, U.: Global organic and inorganic aerosol hygroscopicity and its effect on radiative forcing, Nat. Commun. 2023 141, 14, 6139-, https://doi.org/10.1038/s41467-023-41695-8, 2023.

Ren, R., Li, Z., Yan, P., Wang, Y., Wu, H., Cribb, M., Wang, W., Jin, X., Li, Y., and Zhang, D.: Measurement report: The effect of aerosol chemical composition on light scattering due to the hygroscopic swelling effect, Atmos. Chem. Phys., 21, 9977–9994, https://doi.org/10.5194/ACP-21-9977-2021, 2021.

Rickards, A. M. J., Miles, R. E. H., Davies, J. F., Marshall, F. H., and Reid, J. P.: Measurements of the sensitivity of aerosol hygroscopicity and the κ parameter to the O/C ratio, J. Phys. Chem. A, 117, 14120–14131, https://doi.org/10.1021/JP407991N/SUPPL_FILE/JP407991N_SI_001.PDF, 2013.

Salam, A., Mamoon, H. Al, Ullah, M. B., and Ullah, S. M.: Measurement of the atmospheric aerosol particle size distribution in a highly polluted mega-city in Southeast Asia (Dhaka-Bangladesh), Atmos. Environ., 59, 338–343, https://doi.org/10.1016/j.atmosenv.2012.05.024, 2012.

Savadkoohi, M., Pandolfi, M., Favez, O., Putaud, J. P., Eleftheriadis, K., Fiebig, M., Hopke, P. K., Laj, P., Wiedensohler, A., Alados-Arboledas, L., Bastian, S., Chazeau, B., María, Á. C., Colombi, C., Costabile, F., Green, D. C., Hueglin, C., Liakakou, E., Luoma, K., Listrani, S., Mihalopoulos, N., Marchand, N., Močnik, G., Niemi, J. V., Ondráček, J., Petit, J. E., Rattigan, O. V., Reche, C., Timonen, H., Titos, G., Tremper, A. H., Vratolis, S., Vodička, P., Funes, E. Y., Zíková, N., Harrison, R. M., Petäjä, T., Alastuey, A., and Querol, X.: Recommendations for reporting equivalent black carbon (eBC) mass concentrations based on long-term pan-European

in-situ observations, Environ. Int., 185, 108553, https://doi.org/10.1016/J.ENVINT.2024.108553, 2024.

Savadkoohi, M., Gherras, M., Favez, O., Petit, J. E., Rovira, J., Chen, G. I., Via, M., Platt, S., Aurela, M., Chazeau, B., de Brito, J. F., Riffault, V., Eleftheriadis, K., Flentje, H., Gysel-Beer, M., Hueglin, C., Rigler, M., Gregorič, A., Ivančič, M., Keernik, H., Maasikmets, M., Liakakou, E., Stavroulas, I., Luoma, K., Marchand, N., Mihalopoulos, N., Petäjä, T., Prevot, A. S. H., Daellenbach, K. R., Vodička, P., Timonen, H., Tobler, A., Vasilescu, J., Dandocsi, A., Mbengue, S., Vratolis, S., Zografou, O., Chauvigné, A., Hopke, P. K., Querol, X., Alastuey, A., and Pandolfi, M.: Addressing the advantages and limitations of using Aethalometer data to determine the optimal absorption Ångström exponents (AAEs) values for eBC source apportionment, Atmos. Environ., 349, https://doi.org/10.1016/j.atmosenv.2025.121121, 2025.

Shingler, T., Sorooshian, A., Ortega, A., Crosbie, E., Wonaschütz, A., Perring, A. E., Beyersdorf, A., Ziemba, L., Jimenez, J. L., Campuzano-Jost, P., Mikoviny, T., Wisthaler, A., and Russell, L. M.: Ambient observations of hygroscopic growth factor and f(RH) below 1: Case studies from surface and airborne measurements, J. Geophys. Res. Atmos., 121, 13,661-13,677, https://doi.org/10.1002/2016JD025471, 2016.

Suda, S. R., Petters, M. D., Yeh, G. K., Strollo, C., Matsunaga, A., Faulhaber, A., Ziemann, P. J., Prenni, A. J., Carrico, C. M., Sullivan, R. C., and Kreidenweis, S. M.: Influence of functional groups on organic aerosol cloud condensation nucleus activity, Environ. Sci. Technol., 48, 10182–10190, https://doi.org/10.1021/ES502147Y, 2014.

Titos, G., Lyamani, H., Cazorla, A., Sorribas, M., Foyo-Moreno, I., Wiedensohler, A., and Alados-Arboledas, L.: Study of the relative humidity dependence of aerosol light-scattering in southern Spain, Tellus B Chem. Phys. Meteorol., 66, 24536, https://doi.org/10.3402/tellusb.v66.24536, 2014.

Titos, G., Burgos, M. A., Zieger, P., Alados-Arboledas, L., Baltensperger, U., Jefferson, A., Sherman, J., Weingartner, E., Henzing, B., Luoma, K., O'Dowd, C., Wiedensohler, A., and Andrews, E.: A global study of hygroscopicity-driven light-scattering enhancement in the context of other in situ aerosol optical properties, Atmos. Chem. Phys., 21, 13031–13050, https://doi.org/10.5194/acp-21-13031-2021, 2021.

Wei, F., Peng, X., Cao, L., Tang, M., Feng, N., Huang, X., and He, L.: Characterizing water solubility of fresh and aged secondary organic aerosol in PM2.5 with the stable carbon isotope technique, Atmos. Chem. Phys., 24, 8507–8518, https://doi.org/10.5194/acp-24-8507-2024, 2024.

Wu, T. and Boor, B. E.: Urban aerosol size distributions: A global perspective, Atmos. Chem. Phys., 21, 8883–8914, https://doi.org/10.5194/acp-21-8883-2021, 2021.

Wu, Y., Wang, X., Yan, P., Zhang, L., Tao, J., Liu, X., Tian, P., Han, Z., and Zhang, R.: Investigation of hygroscopic growth effect on aerosol scattering coefficient at a rural site in the southern North China Plain, Sci. Total Environ., 599–600, 76–84, https://doi.org/10.1016/J.SCITOTENV.2017.04.194, 2017.

Zieger, P., Fierz-Schmidhauser, R., Poulain, L., Müller, T., Birmili, W., Spindler, G., Wiedensohler, A., Baltensperger, U., and Weingartner, E.: Influence of water uptake on the aerosol particle light scattering coefficients of the Central European aerosol, Tellus B Chem. Phys. Meteorol., 66, https://doi.org/10.3402/TELLUSB.V66.22716, 2014.